# Wireless recording from unrestrained monkeys reveals motor goal encoding beyond immediate reach in frontoparietal cortex

Michael Berger[1,2]*, Naubahar Shahryar Agha[1], Alexander Gail[1,2,3,4]

[1]Cognitive Neuroscience Laboratory, German Primate Center – Leibniz-Institute for Primate Research, Goettingen, Germany; [2]Faculty of Biology and Psychology, University of Goettingen, Goettingen, Germany; [3]Leibniz-ScienceCampus Primate Cognition, Goettingen, Germany; [4]Bernstein Center for Computational Neuroscience, Goettingen, Germany

**Abstract** System neuroscience of motor cognition regarding the space beyond immediate reach mandates free, yet experimentally controlled movements. We present an experimental environment (Reach Cage) and a versatile visuo-haptic interaction system (*MaCaQuE*) for investigating goal-directed whole-body movements of unrestrained monkeys. Two rhesus monkeys conducted instructed walk-and-reach movements towards targets flexibly positioned in the cage. We tracked 3D multi-joint arm and head movements using markerless motion capture. Movements show small trial-to-trial variability despite being unrestrained. We wirelessly recorded 192 broadband neural signals from three cortical sensorimotor areas simultaneously. Single unit activity is selective for different reach and walk-and-reach movements. Walk-and-reach targets could be decoded from premotor and parietal but not motor cortical activity during movement planning. The Reach Cage allows systems-level sensorimotor neuroscience studies with full-body movements in a configurable 3D spatial setting with unrestrained monkeys. We conclude that the primate frontoparietal network encodes reach goals beyond immediate reach during movement planning.

**\*For correspondence:**
mberger@dpz.eu

**Competing interests:** The authors declare that no competing interests exist.

## Introduction

Cognitive sensorimotor neuroscience investigates how the brain processes sensory information, develops an action plan, and ultimately performs a corresponding action. Experimental setups with non-human primates typically make use of physical restraint, such as a primate chair, to control for spatial parameters like head position, gaze direction, and body and arm posture. This approach led to numerous important insights into neural correlates of visually guided hand and arm movements. The frontoparietal reach network, in particular, including the posterior parietal cortex, premotor cortex, and motor cortex, has been studied in terms of force encoding (*Cheney and Fetz, 1980*), direction encoding (*Georgopoulos et al., 1986*), spatial reference frames of reach goal encoding (*Batista et al., 1999*; *Buneo et al., 2002*; *Kuang et al., 2016*; *Pesaran et al., 2006*), context integration (*Gail and Andersen, 2006*; *Martínez-Vázquez and Gail, 2018*; *Niebergall et al., 2011*; *Westendorff et al., 2010*), obstacle avoidance (*Kaufman et al., 2013*; *Mulliken et al., 2008*), bimanual coordination (*Donchin et al., 1998*; *Mooshagian et al., 2018*), eye-hand coordination (*Hwang et al., 2012*; *Mooshagian and Snyder, 2018*; *Sayegh et al., 2017*; *Wong et al., 2016*), and decision making (*Christopoulos et al., 2015*; *Cisek, 2012*; *Klaes et al., 2011*; *Suriya-Arunroj and Gail, 2019*). Because of the physical restraint, the scope of previous studies was mostly limited to hand or arm movements, and those were restricted to the immediately reachable space.

Well-controlled planning and execution of spatially and temporally structured goal-directed movements in larger workspaces, including reach goals beyond immediate reach, could not be investigated in monkeys.

Neuropsychological and neurophysiological evidence suggest that frontoparietal areas encode the space near the body differently from the space far from the body (see *Farnè et al., 2016* for review). Visuospatial neglect can be restricted to the near or far space as shown by patients with large-scale lesions comprising also parietal cortex (*Halligan and Marshall, 1991*; *Vuilleumier et al., 1998*) and transcranial magnetic stimulation over the parietal cortex (*Bjoertomt et al., 2002*). Bimodal neurons in premotor cortex and the posterior parietal cortex of non-human primates respond to visual and somatosensory stimulation, with visual receptive fields being congruent with somatosensory receptive fields and thereby covering the space near the body (*Colby and Goldberg, 1999*; *Graziano et al., 1997*; *Rizzolatti et al., 1981*; *Rizzolatti et al., 1997*). In addition, mirror neurons in the ventral premotor cortex can respond differently to an observed reach if the reach goal is within its own reach or not. (*Bonini et al., 2014*; *Caggiano et al., 2009*). These findings indicate that encoding of bimodal sensory information and information about observed actions seems to be dependent on one's own body boundaries. Moreover, those findings suggest that premotor and parietal cortex are affected by this distinction. The frontoparietal network encodes motor goals within immediate reach, but it is unclear if this also holds true for motor goals beyond immediate reach. Because of the physical restraint of conventional setups, it has not been possible to investigate naturalistic goal-directed movements that require the monkey to walk towards targets at variable positions and, thus, to investigate how monkeys plan to acquire a reach goal beyond the immediately reachable space.

In conventional experiments, tethered connections prohibit recording from freely moving primates, at least in the case of larger species such as macaques. Tethered recordings in freely moving smaller primate species, such as squirrel monkeys (*Ludvig et al., 2004*) or marmosets (*Courellis et al., 2019*; *Nummela et al., 2017*) have been demonstrated. One study also showed tethered recordings in Japanese macaques; however these were in an environment with no obstacles and with low channel count (*Hazama and Tamura, 2019*). Using wireless recording technology in combination with chronically implanted arrays, recent studies achieved recordings of single unit activity in nonhuman primates investigating vocalization (*Hage and Jurgens, 2006*; *Roy and Wang, 2012*), simple uninstructed behavior (*Schwarz et al., 2014*; *Talakoub et al., 2019*), treadmill locomotion (*Capogrosso et al., 2016*; *Foster et al., 2014*; *Schwarz et al., 2014*; *Yin et al., 2014*), chair-seated translocation (*Rajangam et al., 2016*), sleep (*Yin et al., 2014*; *Zhou et al., 2019*), and simple movements to a food source (*Capogrosso et al., 2016*; *Chestek et al., 2009*; *Fernandez-Leon et al., 2015*; *Hazama and Tamura, 2019*; *Schwarz et al., 2014*; *Shahidi et al., 2019*). An alternative to wireless transmission can be data logging for which the data are stored separately from behavioral data on the headstage (*Zanos et al., 2011*). This has been used in investigations of simple uninstructed behavior and sleep (*Jackson et al., 2006*; *Jackson et al., 2007*; *Xu et al., 2019*). However, none of the experiments with neural recordings in unrestrained monkeys presented an experimental environment that instructs temporally and spatially precise movement behavior (*Supplementary file 1*). To study goal-directed motor planning and spatial encoding of motor goals, we developed the Reach Cage in which we can instruct precise movement start times and multiple distributed movement goals independent from the food source.

Here, we present an experimental environment, the Reach Cage, which is equipped with a visuohaptic interaction system (*MaCaQuE*) and allows investigating movement planning and goal-directed movements of unrestrained rhesus monkeys while recording and analyzing in real-time cortical single-unit activity. We trained monkeys to perform controlled memory-guided reach movements with instructed delay to targets within and beyond the immediately reachable space. Using an open source markerless video-based motion capture software (*Mathis et al., 2018*), we measured 3-dimensional head, shoulder, elbow, and wrist trajectories. We used wireless recording technology to extract single unit activity in real-time from three cortical areas (parietal reach region PRR, dorsal premotor cortex PMd, and primary motor cortex M1) at a bandwidth suitable for brain-machine interface (BMI) applications. We show that the Reach Cage is suitable for sensorimotor neuroscience with physically unrestrained rhesus monkeys providing a richer set of motor tasks, including walk-and-reach movements. With the Reach Cage we were able to study motor goal encoding beyond the immediate reach and during ongoing walking movements. We show that PRR and PMd, but not

M1, already contain target location information of far-located walk-and-reach targets during the planning period before and during the walk-and-reach movement.

## Results

We developed the Reach Cage to expand studies of visual guided reaching movements to larger workspaces and study movements of rhesus monkeys performing structured whole-body movement tasks while being physically unrestrained. We report on quantitative assessment of the animals' behavior in the Reach Cage, and neuroscientific analysis of walk-and-reach goal encoding. The timing of the monkeys' reaching behavior can be precisely controlled and measured with the touch and release times of our touch-sensitive cage-mounted targets (1st section). Additionally, multi-joint 3-dimensional reach kinematics can be measured directly with the video-based motion capture system (2nd section). We will show that high channel count wireless neural recording is possible in the Reach Cage and report on single-unit activity during such structured task performance (3rd section). Finally, we demonstrate the suitability of the Reach Cage for studying motor goal encoding beyond the immediate reach and show that premotor and parietal cortical activity contain information about far-located walk-and-reach targets position during movement planning (4th section).

### Real-time control of instructed behavior in physical unrestrained rhesus monkeys in the Reach Cage

The core element of our newly developed Reach Cage (*Figure 1*) is the *Macaque Cage Query Extension* (*MaCaQuE*). Using this interaction device, we were able to train two fully unrestrained rhesus monkeys to conduct spatially and temporally well-structured memory-guided reaches, a behavioral task common to sensorimotor neuroscience in primates. Here we report the technical details of *MaCaQuE* and its use with physically unrestrained rhesus monkeys; however, we also used *MaCaQuE* successfully in a study with human participants (*Berger et al., 2019*).

Both animals learned within a single first session that touching a target presented on a *MaCaQuE Cue and Target box* (*MCT*, *Figure 1B*) leads to a liquid reward. The computer-controlled precise

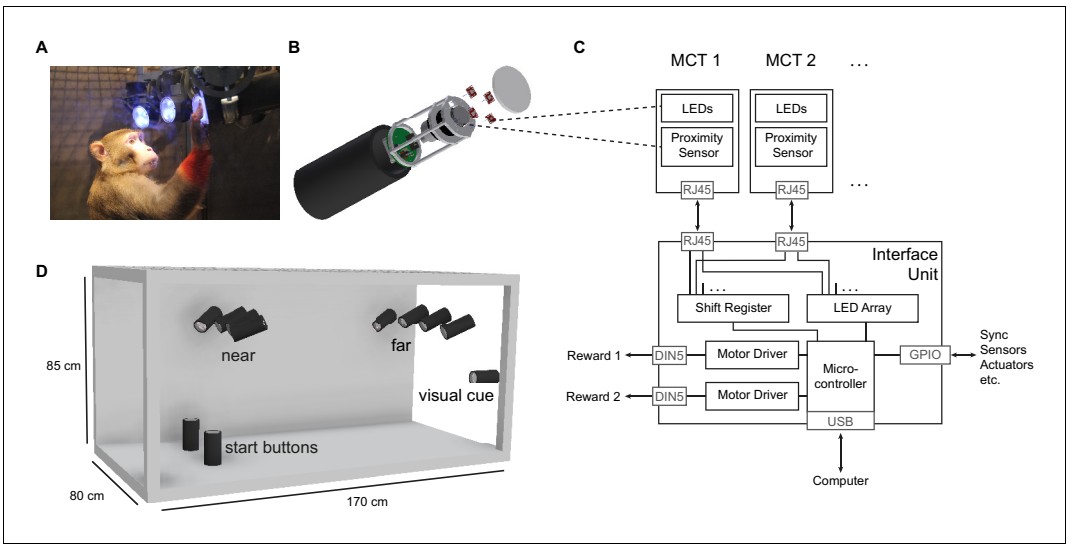

**Figure 1.** The Reach Cage setup. (**A**) Monkey K performing a reach task on the *Macaque Cage Query Extension* (*MaCaQuE*), touching one of the illuminated *MaCaQuE* Cue and Target boxes (*MCTs*) inside the Reach Cage. (**B**) An *MCT* contains a proximity sensor to make the translucent front cover touch-sensitive and four RGB LEDs to color-illuminate it. (**C**) Schematic drawing of *MaCaQuE* showing the electronic components with the microcontroller interfacing between *MCTs* and an external computer for experimental control. (**D**) Sketch of the Reach Cage with 10 *MCTs* inside, two on the floor pointing upwards serving as a starting position for the monkey and two rows of four (near and far) pointing towards the starting position. The far *MCTs* were positioned to the back of the cage such that the animals needed to walk first. An 11th *MCT* is positioned outside the cage for providing additional visual cues. The universal *MCTs* can be arranged flexibly to serve different purposes.

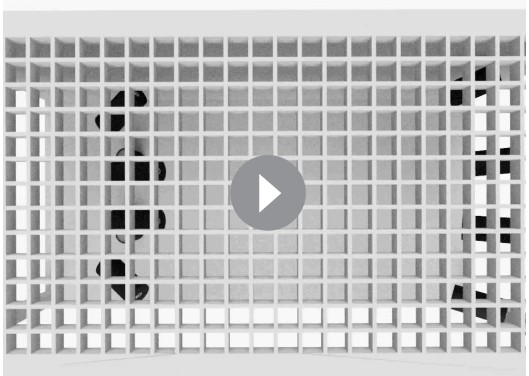

**Video 1.** Three-dimensional animation of the Reach Cage.
https://elifesciences.org/articles/51322#video1

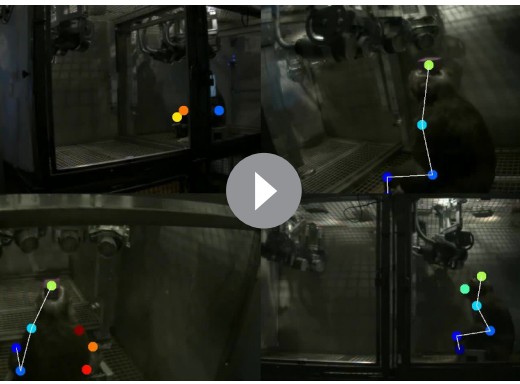

**Video 2.** The video shows reaching movements by monkey K with motion capture labels from all four cameras. One example trial for each near target is depicted.
https://elifesciences.org/articles/51322#video2

timing and dosage of reward (*Figure 1C*) meant that we could employ *MaCaQuE* for positive reinforcement training (PRT) to teach both animals a memory-guided target acquisition task with instructed delay (see Materials and methods). Unlike chair-based setups, *MaCaQuE* allows for target placement beyond the immediate reach of the monkeys (*Figure 1D,Video 1*) Monkey K performed the final version of the walk-and-reach task (*Figure 2A/B*) with 77% correct trials on average (s.d. 9%, 19 sessions) with up to 412 correct trials per session (mean 208, s.d. 93). The sessions lasted on average 40 min (s.d. 15 min). Monkey L performed the final version of the task with 55% correct trials on average (s.d. 5%, 10 sessions) with up to 326 correct trials per session (mean 219, s.d. 55). Sessions lasted on average 65 min (s.d. 15 min). The majority of errors resulted from premature release of the start buttons prior to the go cue. Trials with properly timed movement initiation were 92% correct in monkey K and 78% correct in monkey L.

While the animals were not physically restricted to a specific posture, the strict timing of the task encouraged them to optimize their behavior. As the *MaCaQuE* system makes information about *MCT* touches and releases available with minimal delay (<20 ms), it is possible to enforce an exact timing of the movements when solving a reaching task in the Reach Cage. *Figure 2C* shows the distribution of button release times and movement times towards near and far targets for the task (monkey K/L: 19/10 sessions, 3956/2194 correct trials). As a whole-body translocation is required to

approach far targets, movement times were longer than for near targets in both monkeys (t-test, p<0.001). Movement time distributions were narrow (s.d. ≤ 76 ms), indicating that the monkeys optimized their behavior for consistent target acquisition. Button release time indicates the onset of the hand movement, not necessarily the whole-body movement. In monkey K, the button release times were higher for far compared to near targets (t-test, p<0.001). In contrast, button release times in monkey L were lower for far compared to near targets (p<0.001), reflecting a different behavioral strategy for movement onset (monkey K was sitting during the delay period whereas monkey L was standing).

The behavioral results as directly measured with *MaCaQuE* via the proximity sensors of the *MCTs* demonstrate that the Reach Cage is suitable to train animals on goal-directed reaching

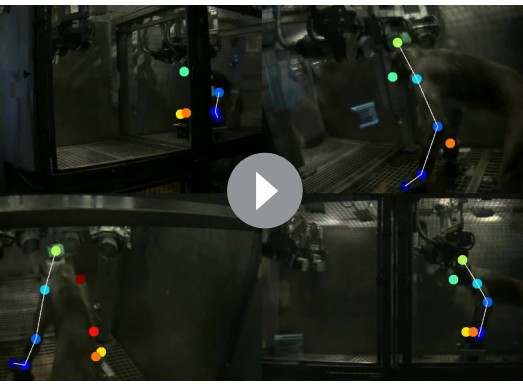

**Video 3.** The video shows reaching movements by monkey L with motion capture labels from all four cameras. One example trial for each near target is depicted.
https://elifesciences.org/articles/51322#video3

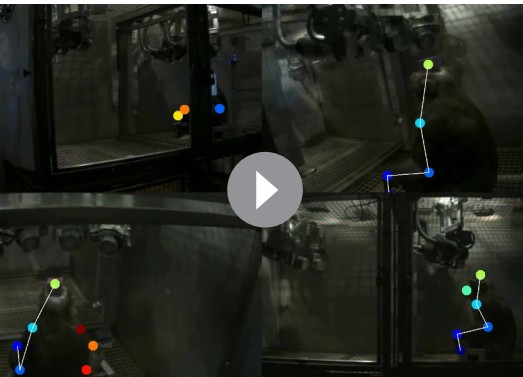

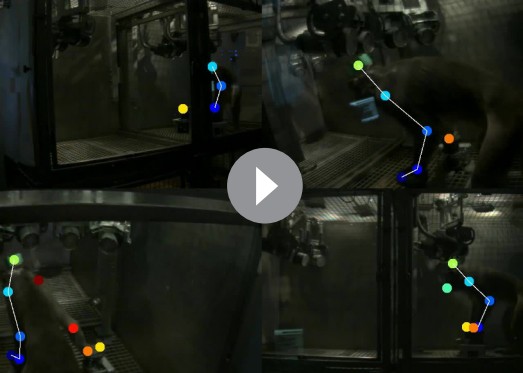

**Video 4.** The video shows walk-and-reach movements by monkey K with motion capture labels from all four cameras. One example trial for each far target is depicted.
https://elifesciences.org/articles/51322#video4

**Video 5.** The video shows walk-and-reach movements by monkey L with motion capture labels from all four cameras. One example trial for each far target is depicted.
https://elifesciences.org/articles/51322#video5

tasks with target positions not being constrained by the immediately reachable space of the animal. The temporally well-structured task performance at the same time allows behavioral and neurophysiological analyses as applied in more conventional settings.

## Time-continuous 3-dimensional arm kinematics during walk-and-reach behavior

As we do not impose physical restraint, the monkeys have more freedom to move than in conventional setups. This allows for testing new behavioral paradigms such as the walk-and-reach task but

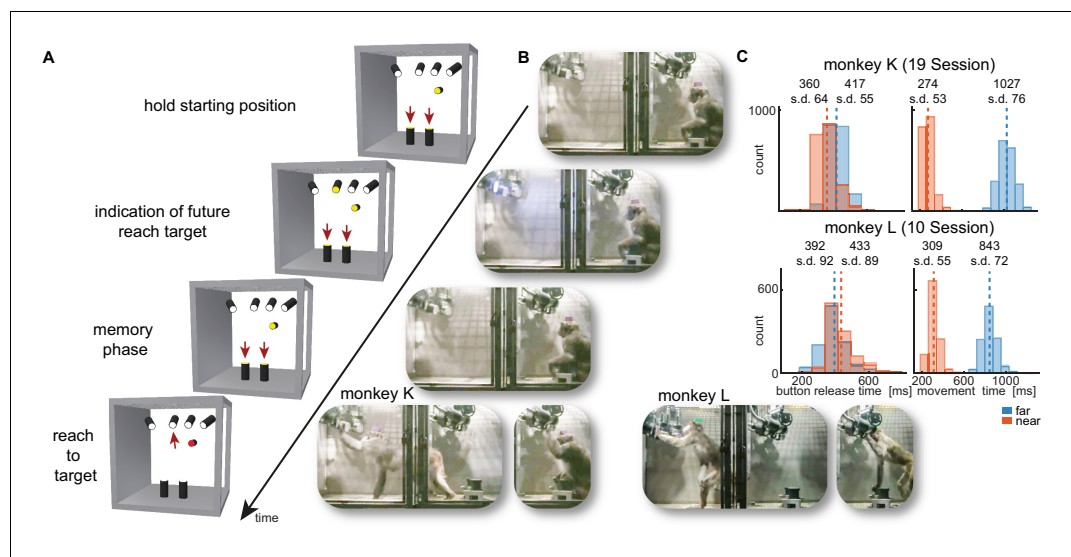

**Figure 2.** Walk-and-reach task. (A) Timeline of the walk-and-reach task. Yellow *MCTs* indicate illumination. Only near targets are shown to illustrate this example trial. The second left-most near target was indicated as target and had to be reached after an instructed memory period in response to the go cue (isoluminant color change on the *MCT* outside the cage). (B) An example trial to a far target for monkey K (left) and monkey L (right). The frames of the video correspond to the time periods of the trial illustrated in (A). (C) Times between go cue and start button release (button release time), and between start button release and target acquisition (movement time) were distributed narrowly in most cases for reaching movements to near (red) and far (blue) targets demonstrating the temporally well-structured behavior. Dashed vertical lines indicate averages and corresponding numbers indicate averages and standard deviations (s.d.) in ms.

also provides more freedom in how to solve the task. We used DeepLabCut (*Mathis et al., 2018*) for video-based motion capture and analyzed kinematics and their variability.

We measured the 3-dimensional posture of the reaching arm during the reach and walk-and-reach behavior of 2/3 sessions with a total of 469/872 successful trials in monkey K/L. Specifically, we tracked the monkeys' headcap, left shoulder, elbow, and wrist (*Videos 2–5*). *Figure 3A* shows the side-view of the body part positions for each trial and video frame between 100 ms before button release and 100 ms after target acquisition for the reach (red) and walk-and-reach (blue) movements to the mid-left target.

Within each animal, reach kinematics were highly consistent from trial to trial and from session to session. To quantify the variability in arm posture, we calculated for each target and marker separately and at corresponding times the Euclidean distance between the single-trial trajectories and the across sessions trial-averaged trajectory. *Figure 3B* shows the distributions of Euclidean distance averaged over time for each trial, marker, and monkey. The highest variability was seen in the wrist during walk-and-reach movements with a median of 37/46 mm and 0.75-quartile of 50/67 mm for monkeys K and L, respectively. Within a session these median deviations are 1–6 mm smaller. As a reference, the transparent front plate of the targets has a diameter of 75 mm. The center-to-center distance between neighboring targets is around 130 mm (near; shown as dashed line in the plot) and 210 mm (far). This shows that even across sessions, the arm posture during the movements towards the same target at a given time varied only by a few centimeters.

The movement patterns between monkey K (left) and monkey L (right) were different. *Figure 3C* shows the trial-averaged arm posture for each time point during the reach and walk-and-reach

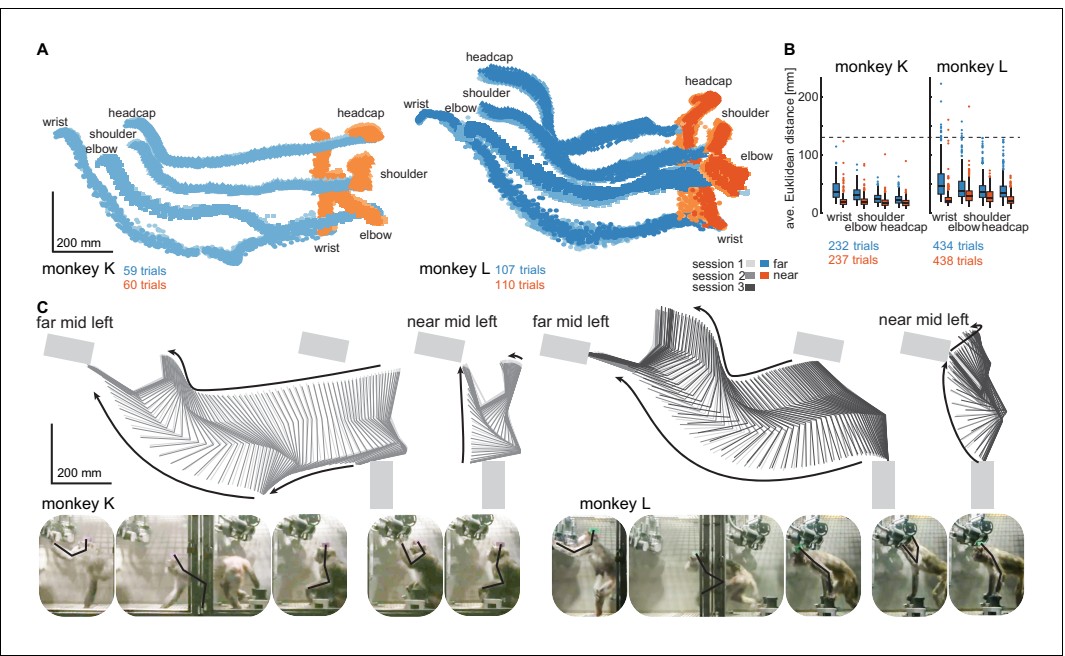

**Figure 3.** Structured behavior during task performance in unrestrained animals. (**A**) Motion tracking of the left wrist, elbow, shoulder, and the headcap implant during reach and walk-and-reach movements for monkey K (left) and monkey L (right). Video-based markers are tracked in three dimensions and projected to a side-view. Trial-by-trial marker positions for the reach (red) and walk-and-reach (blue) movements to the mid-left targets are shown for a sampling frequency of 60 Hz, overlaid for multiple sessions (light-dark colors). (**B**) Small trial-to-trial variability of movement trajectories, even across sessions, demonstrates spatially well-structured and consistent behavior. For each trial and marker, the average Euclidean distance to the trial-averaged trajectory at corresponding times is shown (see Materials and methods). For reference, neighboring near targets were mounted at approximately 130 mm distance (dashed line) in this experiment. The *MCT* diameter is 75 mm. (**C**) Reconstructed 3-dimensional arm posture as function of time during reach and walk-and-reach movements based on the video motion capture separately for each monkey and session. The lines connect the marker (wrist to elbow to shoulder to headcap) for each marker position averaged across trials. Gray rectangles show target and start button *MCTs*. Pictures below show snapshots of characteristic postures during an example reach and walk-and-reach trial.

movements. Monkey K was sitting during the memory period and then used its left forelimb for walking and reaching. Monkey L was standing during the memory period and walked bipedally to the far targets. Both animals used this strategy consistently in all trials.

The kinematic analyses demonstrate that the animals not only complied with the spatial and temporal task requirements in terms of starting and endpoint acquisition but also adopted reliable repetitive behavior in terms of overall reach kinematics despite the lack of physical restraint. The animals used different behavioral strategies. However, the video-based motion capture allowed us to quantify the arm and head kinematics.

## Multi-channel single unit activity can be recorded in the Reach Cage using wireless technology

The Reach Cage provides an adequate setting for studying well-isolated single neuron activity from multiple areas of the cerebral cortex of monkeys during movement planning and execution of goal-directed behavior in minimally constrained settings. We here provide simultaneous recordings from three different sensorimotor areas, including non-surface areas inside sulci, during the goal-directed memory-guided walk-and-reach task.

We chronically implanted a total of 192 electrodes in primary motor cortex (M1), dorsal premotor cortex (PMd), and the parietal reach region (PRR) in the posterior parietal cortex of both monkeys using six 32-channel floating microwire arrays (FMA) with various lengths (see Materials and methods). We recorded broadband (30 ksps per channel) neural data from all arrays simultaneously (i.e.

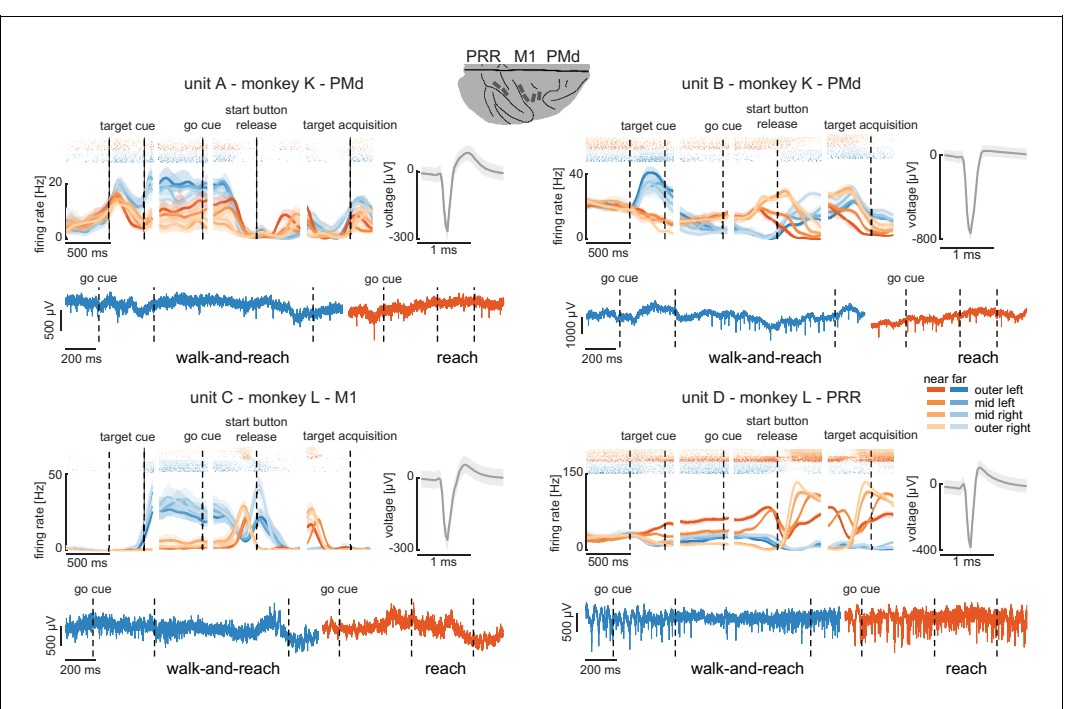

**Figure 4.** Wireless recording in the Reach Cage. Four example units from the frontoparietal reach network of monkeys K and L recorded wirelessly while the monkeys performed the memory-guided walk-and-reach task. The figure shows for each unit averaged spike densities with corresponding raster plots (top left), the waveform (top right), and the unfiltered broadband signal during reach and walk-and-reach example movements. Vertical dashed lines indicate task events in order of appearance: target cue (on and off), go cue, start button release, and target acquisition. Error bars indicate bootstrapped 95% confidence interval for the spike densities and s.d. for the waveform. Color indicates near (red) and far (blue) targets, lightness level indicates right (light) to left (dark) target positions.

The online version of this article includes the following source data and figure supplement(s) for figure 4:

**Source data 1.** Data loss rate differences across targets for all trials.
**Source data 2.** Data loss rate differences across targets for trials below 5% data loss.
**Figure supplement 1.** Data loss rate per target.

up to 192 channels) while the monkeys performed the walk-and-reach task (*Figure 4*). The animals moved through the cage wearing the wireless electronics and protective cap without banging the implant into the cage equipment and performed the behavioral task as without the wireless gear.

We recorded in monkey K/L 2/10 sessions from all six arrays simultaneously using two 96-channel wireless headstages. Our custom-designed implants can house both headstages and protect them and the array connectors against dirt and physical impact. The implants are designed to be used with different commercially available wireless systems, with the 2 × 96 channel digital systems presented here or with a 31- or 127-channel analog wireless system, dependent on the need of the experiment. Implant development and methodological details will be discussed below (Material and methods).

The wireless signal transmission was stable during walking movements. To quantify the stability, we calculated the rate of data loss from lost connection to the wireless system. We checked for each time point if either of the two headstages did not receive data. As a conservative measure, we only considered correctly performed trials, as in these trials it is guaranteed that the animal moved the full stretch from start to goal. The best sessions showed loss rates of 3.18%/1.03% of all time bins for monkey K/L, and worst sessions of 6.59%/6.34%, respectively. On average across sessions and monkeys, the loss rate was 3.32% (s.d. 1.7%). Data loss was spread over all targets with a slight spatial bias (*Figure 4—figure supplement 1A* and *Figure 4—source data 1*, 2 -way ANOVA position $F_{(3, 2657)}=3.48$, $p=0.015$; position x distance $F_{(3, 2657)}=4.81$, $p=0.002$). The spatial bias was introduced by trials with high data loss rates. When removing trials with a loss rate of above 5% there was no significant spatial bias anymore (*Figure 4—figure supplement 1B* and *Figure 4—source data 2*, 2-way ANOVA position $F_{(3, 2657)}=0.88$, $p=0.45$; position x distance $F_{(3, 2657)}=2.36$, $p=0.07$). From here on, we only consider correct trials with a loss rate of less than 5%. Note, walk-and-reach trials showed different loss rates from reach trials ($F_{(3, 2657)}=279.96$, $p<0.001$); however, this does not influence further results that focus on movement direction of reach or walk-and-reach movements separately.

The wireless signal quality was stable during walking movements and allowed us to isolate single- and multi-unit activity during the walk-and-reach task. *Figure 4* shows four example neurons from the frontoparietal reach network of both monkeys while performing the task. Trial-averaged spike densities (top left) show that units were modulated by task condition. All four example neurons are significantly modulated by target distance, left-to-right target position, time during the trial, and interactions of distance x position and distance x time (ANOVA $p<0.05$). Units A and C are mostly active during the memory period while units B and D are active during memory period and movement. Waveforms of the isolated example neurons are shown on the top right of each panel. Unfiltered broadband data of one near (red) and one far (blue) example trial are shown below. Spiking activity can be identified in the broadband signal also during the reach and walk-and-reach movement. We performed the same ANOVA for the activity in each channel of all 12 recorded sessions. Three sessions revealed task-responsive activity on all 192 channels, that is showed at least one effect in distance, position, time, or one of the interactions; across all sessions the mean number of task-responsive channels was 189 (s.d. five channels). Up to 179 channels were position-responsive, that is showing at least one effect in position or one of the interactions (mean: 162, s.d. 17 channels).

In summary, the Reach Cage proved to be suitable for addressing neuroscientific questions based on single and multi-unit recordings. Broadband wireless neural signals showed excellent spike isolation and modulation of spike frequency correlated with behavioral events.

## Premotor and parietal cortex encode movement goals beyond immediate reach

The Reach Cage allows us for the first time to test the spatial encoding of movement goals at larger distances to the animal. We wanted to know whether the frontoparietal reach network encodes motor goals only within the immediate reach or also beyond. For this, we computed separately in near and far trials the performance for decoding goal direction (left vs. right) with a support vector machine (SVM) decoder based on multi-unit firing rates.

We analyzed the session with the highest number of trials for each animal to avoid biasing our results by reusing repeated measures of the same neurons on channels that showed stable signals across multiple sessions. *Figure 5A* shows the movement paths of the wrist (top) and head (bottom)

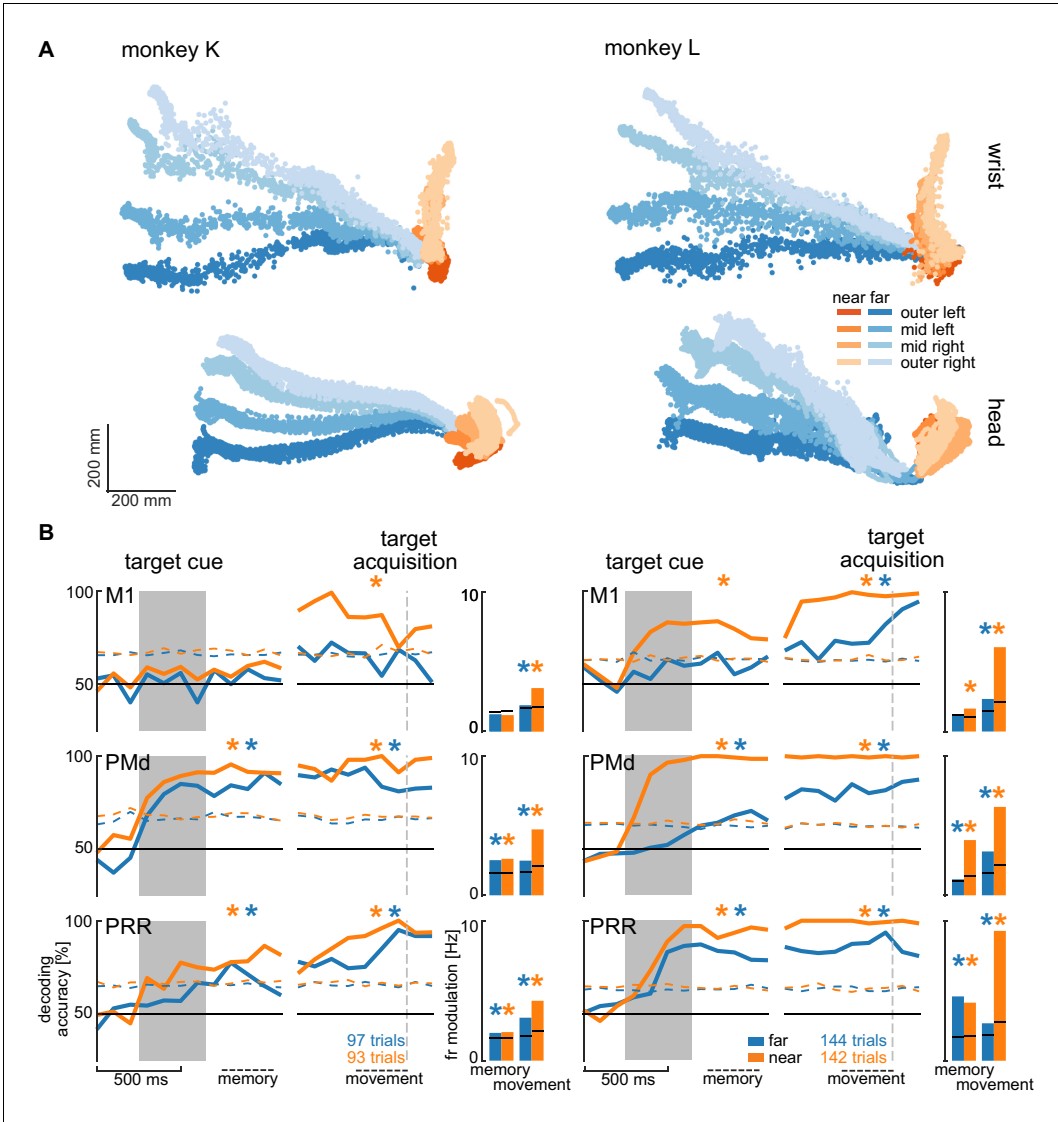

**Figure 5.** Direction decoding in the walk-and-reach task. (**A**) Wrist (top) and head (bottom) position during reach (orange) and walk-and-reach (blue) movements towards the eight targets projected to the top-view. Each point corresponds to one location in one trial sampled at 60 Hz. (**B**) Decoding accuracy of 20-fold cross validation of a linear SVM decoder in 300 ms bins at 100 ms time steps (line plots). We decoded if a trial was towards one of the two left or one of the two right targets. Premotor and parietal cortex but not motor cortex showed significant decoding walk-and-reach targets even during the memory period. Statistical testing was done on one bin in the memory period 100–400 ms after the cue and movement period 300–0 ms before target acquisition (black dashed line). The colored dashed curve indicates the significance threshold based on a one-tailed permutation test. The population average of the firing rate modulation between preferred and anti-preferred direction (left vs. right) during the memory and movement bin is shown in the bar plots. The black lines indicate the significance threshold based on a one-tailed permutation test. In both plots, an asterisk corresponds to a significant increase with Bonferroni correction.

The online version of this article includes the following source data and figure supplement(s) for figure 5:

**Source data 1.** Test for significant SVM decoding accuracy.
**Source data 2.** Test for significant firing rate modulation.
**Source data 3.** Test for change in decoding accuracy between trials with and without passage.
**Source data 4.** Test for change in firing rate modulation between trials with and without passage.
**Figure supplement 1.** Decoding walk-and-reach goals with different walking paths.

of the animals for the reach (orange) and walk-and-reach (blue) behavior towards the different targets. *Figure 5B* (line plots) shows 20-fold cross validation of decoding accuracy in 300 ms time windows at 100 ms time steps. To test if there is reach goal encoding during movement planning prior to onset of movement, we analyzed the time window during the memory period starting 100 ms after target cue offset. To test if there is reach goal encoding during reaching (near) and during ongoing walking-and-reaching (far), we analyzed the 300 ms immediately before target acquisition. We performed a one-tailed permutation test to determine whether decoding accuracy is significantly above chance. In PMd and PRR, decoding is significant for both memory and movement period for reach and walk-and-reach movements (*Figure 5—source data 1*). In M1, decoding accuracy only reached significance for walk-and-reach movements in monkey L during the movement period. We then analyzed the population average of the firing rate modulation between the preferred and anti-preferred direction, again left vs. right (*Figure 5B* bar plots). We performed a one-tailed permutation test to determine significance. As in the decoding analysis, PMd and PRR show significant firing rate modulation for both memory and movement period for reach and walk-and-reach movements (*Figure 5—source data 2*). Here, M1 shows significant firing rate modulation during the walk-and-reach movement period for both monkeys but, as for the decoding analysis, not during the memory period.

From the horizontal fanning out of the unconstrained movement patterns (*Figure 5A*), it became evident that both animals directed their walking movement towards the goal from early on in the movement. To confirm that the motor goal information decodable from PMd and PRR correlates with the reach goal location rather than initial walking movement direction, we introduced a passage in the middle of the walk-and-reach path (a transparent divider between near and far targets with a narrow opening cut out). Although movement trajectories for the different motor goal locations collapsed onto very similar initial walking directions because of the passage (*Figure 5—figure supplement 1A*), the decoding accuracy and firing rate modulation was not affected by this measure, that is was independent of the movement path (*Figure 5 – figure supplement 1B* and *Figure 5—source data 3* and *4*).

Taken together, the Reach Cage environment allows us to study sensorimotor neuroscience questions within an unrestrained spatial setting. Here, we show that target location information is present in premotor and parietal cortex of far-located targets beyond the immediate reach.

## Discussion

We introduced the Reach Cage as a novel experimental environment for sensorimotor neuroscience with physically unrestrained rhesus monkeys. As a core interactive element, we developed *MaCaQuE*, a new experimental control system for sensorimotor tasks in cage environments. We trained two monkeys to conduct spatially and temporally structured memory-guided reach tasks that required them to reach to targets near or far from them with a walk-and-reach movement. With *MaCaQuE*, we could measure button release and movement times in response to visual cues with the same if not higher temporal precision as in touch screen experiments. Using markerless video-based motion capture, we could track 3-dimensional head and multi-joint arm kinematics for reach and walk-and-reach movements. Trajectories had low spatial variability over trials, showing that monkeys perform instructed movement consistently even when no physical restraint is applied. Variations in movement pattern between task conditions or monkeys could well be quantified in detail with this motion capture approach. In parallel, we wirelessly recorded broadband neural signals of 192 channels from three brain areas (M1, PMd, and PRR) simultaneously, an approach suitable for BMI applications. Isolated single-neuron activities were clearly modulated by the task events and encoded information about the location of immediate reach targets and also of remote walk-and-reach targets. Moreover, we could identify walk-and-reach target location information in premotor and parietal cortex, but not motor cortex, during movement and even during the memory period before the movement. This suggests that premotor and parietal cortex encodes motor goals beyond immediate reach. With our Reach Cage approach, we provide an experimental environment that allows testing fully unrestrained monkeys on spatially and temporally controlled behavior. With wireless intra-cortical recordings and markerless motion capture experimental spatial configurations are possible that are not restricted to the vicinity of the animals but allow studying complex full-body movement patterns.

## Far-space motor goal encoding in the frontoparietal reach network

We showed that during the memory period of the walk-and-reach task, target location information of near-located reach and far-located walk-and-reach trials was present in PRR and PMd. Reducing the initial walk-and-reach movement path to a minimum variability between the different target directions by introducing a passage did not change decoding accuracy. This indicates that PRR and PMd activity contains spatial information about the reach goal beyond the immediate reach.

PMd (e.g. *Crammond and Kalaska, 1994*) and PRR (e.g. *Snyder et al., 1998*) activity are known to encode reach related spatial information during planning of reaches within immediate reach. We now show that this is also true beyond reach when walking behavior is needed to approach the reach target. Monkey K even used its reaching arm for walking by making ground contact, while monkey L was swinging its reaching arm during the locomotion without putting it down. This result might seem surprising in view of 1) neuropsychological studies showing that a near space specific neglect can arise from parietal lesions (*Halligan and Marshall, 1991*; *Vuilleumier et al., 1998*) or parietal transcranial stimulation (*Bjoertomt et al., 2002*) and 2) the existence of bimodal neurons in premotor and posterior parietal cortex that have visual receptive fields centered on body surface and only covering its vicinity (*Colby and Goldberg, 1999*; *Graziano et al., 1997*; *Rizzolatti et al., 1981*; *Rizzolatti et al., 1997*). Yet, none of these studies explicitly show nor disregard PMd or PRR being involved in far space encoding. It could be, for example, that such a near or far space specificity is located in separate parts of premotor or parietal cortex. However, we propose an alternative explanation. The extent of the near space, often called peripersonal space (*Rizzolatti et al., 1997*), is variable. Neurophysiological and neuropsychological studies have shown that it can expand around tools (*Berti and Frassinetti, 2000*; *Giglia et al., 2015*; *Holmes, 2012*; *Iriki et al., 1996*; *Maravita et al., 2002*; *Maravita and Iriki, 2004*) or fake arms (*Blanke et al., 2015*; *Botvinick and Cohen, 1998*; *Graziano et al., 2000*; *Maravita et al., 2003*; *Pavani et al., 2000*). There is evidence from human psychophysics that the peripersonal space, here defined by the spatial extent of visuo-tactile integration, expands towards reach goals (*Brozzoli et al., 2009*; *Brozzoli et al., 2010*). Correspondingly, we could show that peripersonal space, as defined by the occurrence of visuo-tactile integration, in human participants expands to reach goals beyond immediate reach when subjects performed a walk-and-reach task similar to here (*Berger et al., 2019*). While previous research suggested selective encoding of near space in parts of parietal and premotor cortex, goal-directed behavior might lead to an expansion of so-called near space even beyond immediate reach. Far-located walk-and-reach goals hence might effectively be within the 'near space' and be encoded similar to near-located reach goals in parietal and premotor regions known for reach goal selectivity during planning and movement.

## Neuroscience of goal-directed behavior in unrestrained non-human primates

As the example of far-space encoding above demonstrates, our understanding of motor cognition and spatial cognition in the primate brain might underestimate the true complexity of cortical representations as experimental needs previously prevented the study of more involved goal-directed full-body movements. While the limitations imposed by tethered recording techniques have been overcome with wireless technologies or data-logging in several neurophysiological studies with unrestrained non-human primates, the investigation of sensorimotor behavior has so far mostly focused on locomotion behavior, such as treadmill or corridor walking, or immediate collection of food items with the forelimb (see *Supplementary file 1* for an overview). In none of these previous studies, was precisely timed and spatially well-structured goal-directed behavior, or even movement planning, investigated in unrestrained monkeys. If behavior was 'instructed', it was always a direct movement towards a food source. Our Reach Cage made it possible to have multiple movement targets dislocated from the food source and placed at variable locations within the cage. Also, it allowed provision of strict temporal instructions to the animals on when to start or when to finish a movement.

With the Reach Cage we aimed for an experimental setting which allows us to study spatial cognitive and full-body sensorimotor behavior with levels of experimental control and behavioral analysis equivalent to conventional chair-seated experiments. We aimed for maximal freedom of the animal to move, and combined this with the conventional approach of a highly trained and structured task

that (1) allows us to control movement timing to introduce certain periods, such as movement preparation; (2) ensures that the animal focuses on the specific behavior required by the task demand; and (3) provides repetition for a statistical analysis. With this combination, we were able to train the animals to conduct goal-directed memory-guided walk-and-reach movements upon instruction, a behavior which cannot be studied in chair-based settings or on treadmills.

The animals' movement behavior was only constrained by the task and the overall cage volume. Nonetheless, reach trajectories revealed fast straight movements with little trial-to-trial variability even across sessions. Apparently, over the course of training, the animals had optimized their movement behavior and adopted consistent starting postures and stereotyped movement sequences. We were able to use the interaction device *MaCaQuE* to reveal narrow distributions of hand release time of the start button as response to the go signal and the movement time from the start button to the reach target. This spatiotemporal consistency of the behavior over many trials allows analytical approaches to both the behavioral and the neural data equivalent to conventional settings.

*MaCaQuE* can serve as a robust cage-based equivalent to illuminated push-buttons (*Batista et al., 1999*; *Buneo and Andersen, 2012*) or a touch screen (*Klaes et al., 2011*; *Westendorff et al., 2010*) in conventional experiments, or as an alternative to wall-mounted touch screens in the housing environment (*Berger et al., 2018*; *Calapai et al., 2017*). Yet, the *MaCaQuE* system is more flexible in terms of spatial configuration. Targets and cues are vandalism-proof and can be placed at any position in large enclosures, allowing for 3-dimensional arrangements and an arbitrarily large workspace. If more explorative, less stereotyped behavior is of interest, the trial-repetitive nature of the current task can easily be replaced by alternative stimulus and reward protocols, for example for foraging tasks. Our reach goal decoding analysis performed on a single trial basis showed that single trial quantification is possible. This would allow for the analyses of unstructured behavior. In another study, we used *MaCaQuE* with humans and expanded it to deliver vibrotactile stimuli to the subjects' fingers and to receive additional input from push buttons in parallel to the reach target input and output (*Berger et al., 2019*). It would also be straightforward to implement continuous interaction devices such as a joystick or motors to control parts of the cage, for instance doors. Similar to other systems for neuroscience experimentation and training (*Libey and Fetz, 2017*; *Ponce et al., 2016*; *Teikari et al., 2012*), we used low-cost off-the-shelf components with an easy-to-program microcontroller platform as a core.

## Wireless recordings for BMI applications

An important translational goal of sensorimotor neuroscience with non-human primates is the development of BMI based on intracortical extracellular recordings to aid patients with severe motor impairments. Intracortical signals can be decoded to control external devices, as demonstrated in non-human primates (*Carmena, 2013*; *Hauschild et al., 2012*; *Musallam et al., 2004*; *Santhanam et al., 2006*; *Serruya et al., 2002*; *Taylor, 2002*; *Velliste et al., 2008*; *Wessberg et al., 2000*), and suited to partially restore motor function in quadriplegic human patients (*Aflalo et al., 2015*; *Bouton et al., 2016*; *Collinger et al., 2013*; *Gilja et al., 2015*; *Hochberg et al., 2012*; *Wodlinger et al., 2015*). The results from the Reach Cage allow relevant insight towards BMI applications in two ways. First, we show encoding of reach goals during other ongoing movement behavior (locomotion). A previous study showed that when monkeys perform an arm movement task in parallel to a BMI cursor task based on decoding arm movement related neural activity, the BMI performance decreases (*Orsborn et al., 2014*). Little was known before about the stability of forelimb decoding performance when other body movements are performed in parallel, such as walking. For partially movement-impaired patients, such as arm amputees, existence of reach goal signals, as demonstrated here, is a prerequisite for restoring the lost function with a prosthesis while still conducting the healthy movements, for example walking. Second, the Reach Cage in its current form with its discrete lights and targets provides a useful environment for BMI studies that follow a different approach, namely to control smart devices or a smart home with ambient assisted living environments reacting to discrete sets of commands. While the user only needs to choose among a discrete set of programs, the smart device or home would take care of the continuous control of the addressed actuators. The Reach Cage is a useful tool to develop such a BMI that makes temporally precise and correct decisions on which program to activate. Importantly, the Reach Cage allows to test if, and in which brain areas, such decisions are encoded invariant to body position in the room,

important also for patients incapable of walking but using assistdevices such as a wheelchair to relocate (*Rajangam et al., 2016*).

We show that our recording bandwidth and quality is sufficiently high for analyzing neural spiking activity in multiple brain areas simultaneously. Further, we show that there is enough information in the population activity to be detected by a decoder on a single trial basis. This is an important prerequisite for BMI applications, and also for the analysis of free behavior, for which structured repetitive behavior is neither given nor wanted. To our knowledge, 192 channels is the highest channel count of recording full broadband (30 ksps per channel) intracortical recordings in unrestrained non-human primates. Previous studies presented simultaneous recordings of 96 channels broadband data; when higher channel counts were used, for example spiking activity from 512 channels (*Schwarz et al., 2014*), automatic spike detection on the headstage was applied and only spike times and waveforms were transmitted and recorded. This is sufficient for spike time analyses but full broadband data would be necessary to extract local field potentials and to change spike detection post-hoc.

An alternative to wireless recordings is data logging, which can be used to store the recorded data on a head-mounted device (*Jackson et al., 2006*; *Jackson et al., 2007*; *Zanos et al., 2011*). While the logging device is detached from any behavioral monitoring or task instruction system, additional measures can be taken to ensure offline synchronization of behavioral data with the logged neural data. Yet, real-time spike sorting and data processing for closed-loop BMI applications are limited to the head-mounted computational capacity when using loggers, which is usually low, while a wireless transmission provides access to powerful processors outside the animal.

## Three-dimensional markerless motion capture in the Reach Cage

In addition to *MaCaQuE* for experimental control, we demonstrated the usefulness of 3-dimensional video-based multi-joint motion tracking during the walk-and-reach movements. Reliable motion capture with unrestrained monkeys provides a technical challenge. At least two cameras need to see a marker or body part to reconstruct a 3-dimensional position. Occlusion by objects or the animal itself is usually an issue (*Chen and Davis, 2000*; *Moeslund et al., 2006*). When using systems based on physical markers (active LEDs or passive reflectors), rhesus monkeys tend to rip off the markers attached to their body, unless excessively trained. An alternative is fluorescent or reflective markers directly painted to the skin of the animal (*Courtine et al., 2005*; *Peikon et al., 2009*), which also require continuously repeated shaving, or markers that cannot be removed, such as collars (*Ballesta et al., 2014*). Video-based marker-free system models designed for use with monkeys were recently reported (*Bala et al., 2020*; *Nakamura et al., 2016*); however, these are not yet reported with neurophysiological recordings. We used the recently introduced open source toolbox DeepLabCut (*Mathis et al., 2018*), which provides markerless tracking of visual features in a video, such as body parts but also objects. DeepLabCut provides excellent tracking of body parts from different species such as monkeys (*Labuguen et al., 2019*), mice, flies, humans, fish, horses, and cheetahs (*Nath et al., 2019*), While we focus on instructed behavior, the current motion capture setting would allow quantification of 3-dimensional free behavior of non-human primates given an appropriate number of camera views.

## Conclusion

Systems neuroscience can benefit substantially from the possibility of quantifying free behavior and simultaneously recording large-scale brain activity, particularly, but not only in sensorimotor research. This possibility opens a range of new opportunities, for example to study motor control of multi-joint and whole-body movements, spatial cognition in complex workspaces, or social interactive behavior. With the opportunities that wireless technology offers, a desirable approach would be to let the monkey freely decide on its behavior to obtain neural correlates of most natural behavior (*Gilja et al., 2010*) while motion capture provides the related movement kinematics (*Bala et al., 2020*; *Ballesta et al., 2014*; *Bansal et al., 2012*; *Mathis et al., 2018*; *Nakamura et al., 2016*; *Peikon et al., 2009*). In fact, we consider it an important next step in systems neuroscience to demonstrate that the important and detailed knowledge that has been gained from tightly controlled experimental settings generalizes well to more naturalistic behaviors. Here, with the Reach Cage we present an experimental environment in combination with high-channel count wireless recording

from multiple brain areas and with multi-joint markerless motion capture. We demonstrated that we can use this setting to study instructed behavior, for which it is easier to isolate different behavioral aspects of interest (movement planning, walking, and reaching). This allowed us to isolate movement planning related activity to reach targets outside of the immediate reach. We could show that the frontoparietal reach network encodes such far-located reach goals.

## Materials and methods

### Animals

Two male rhesus monkeys (Macaca mulatta K age: 6 years; and L age: 15 years) were trained in the Reach Cage. Both animals were behaviorally trained with positive reinforcement learning to sit in a primate chair. Monkey K did not participate in any research study before but was trained on a goal-directed reaching task on a touch screen device in the home enclosure (*Berger et al., 2018*). Monkey L was experienced with goal-directed reaching on a touch screen and with a haptic manipulandum in a conventional chair-seated setting before entering the study (*Morel et al., 2016*). Both monkeys were chronically implanted with a transcutaneous titanium head post, the base of which consisted of four legs custom-fit to the surface of the skull. The animals were trained to tolerate periods of head fixation, during which we mounted equipment for multi-channel wireless recordings. We implanted six 32-channel floating microelectrode arrays (Microprobes for Life Science, Gaithersburg, Maryland) with custom electrode lengths in three areas in the right hemisphere of cerebral cortex. Custom-designed implants protected electrode connectors and recording equipment. The implant design and implantation procedures are described below.

Both animals were housed in social groups with one (monkey L) or two (monkey K) male conspecifics in facilities of the German Primate Center. The facilities provide cage sizes exceeding the requirements by German and European regulations, and access to an enriched environment including wooden structures and various toys (*Calapai et al., 2017*). All procedures have been approved by the responsible regional government office [Niedersächsisches Landesamt für Verbraucherschutz und Lebensmittelsicherheit (LAVES)] under permit numbers 3392 42502-04-13/1100 and comply with German Law and the European Directive 2010/63/EU regulating use of animals in research.

### MaCaQuE

We developed the *Macaque Cage Query Extension* (*MaCaQuE*) to provide computer-controlled visual cues and reach targets at freely selectable individual positions in a monkey cage (*Figure 1*). *MaCaQuE* comprises a microcontroller-based interface, controlled via a standard PC, plus a variable number of *MaCaQuE* Cue and Target boxes (*MCT*).

The *MCT* cylinder is made of PVC plastic and has a diameter of 75 mm and a length of 160 mm. At one end of the cylinder the *MCTs* contain a capacitive proximity sensor (EC3016NPAPL, Carlo Gavazzi, Steinhausen, Switzerland) and four RGB-LEDs (WS2812B, Worldsemi Co., Daling Village, China), both protected behind a clear polycarbonate cover. With the LEDs, light stimuli of different color (8-bit color resolution) and intensity can be presented to serve as visual cues (*Figure 1B*). The LEDs surround the proximity sensor, which registers when the monkey touches the middle of the polycarbonate plate with at least one finger. This way the *MCT* acts as a reach target. LEDs, sensor plus a custom-printed circuit board for controlling electronics and connectors are mounted to a custom-designed 3D-printed frame made out of PA2200 (Shapeways, New York City, New York). A robust and lockable RJ45 connector (etherCON, Neutrik AG, Schaan, Liechtenstein) connects the *MCT* to the interface unit from the opposite side of the cylinder via standard Ethernet cables mechanically protected inside flexible metal tubing. The RGB-LEDs require an 800 kHz digital data signal. For noise reduction, we transmit the signal with a differential line driver (SN75174N, SN74HCT245N, Texas Instruments Inc, Dallas, Texas) via twisted-pair cabling in the Ethernet cable to a differential bus transreceiver (SN75176B, Texas Instruments Inc) on the *MCT*. Ethernet cables are CAT 6; however, any other category would be suitable (CAT 1 up to 1 MHz). This setting allows us to use cables at least up to 15 m. Hence, there are no practical limits on the spatial separation between *MCTs* and from the interface for applications even in larger animal enclosures. We did not test longer cables. Apart from the one twisted-pair for the data stream of the RGB-LEDs, the Ethernet cable transmits 12 V power from the interface unit and the digital touch signal from the

proximity sensor to the interface unit. The proximity sensor is directly powered by the 12 V line. The LEDs receive 5 V power from a voltage regulator (L7805CV, STMicroelectronics, Geneva, Switzerland) that scales the 12 V signal down. The PVC and polycarbonate enclosure of the *MCT* as well as the metal cable protection are built robustly enough to be placed inside a rhesus monkey cage. *MaCaQuE* incorporates up to two units to deliver precise fluid rewards (*Calapai et al., 2017*). Each unit consists of a fluid container and a peristaltic pump (OEM M025 DC, Verderflex, Castleford, UK). MOSFET-transistors (BUZ11, Fairchild Semiconductor, Sunnyvale, California) on the interface unit drive the pumps.

The *MCTs* and reward systems are controlled by the Arduino-compatible microcontroller (Teensy 3.x, PJRC, Sherwood, Oregon) placed on a custom-printed circuit board inside the interface unit (*Figure 1C*). To operate a high number of *MCTs* the microcontroller communicates with the proximity sensor and LEDs using two serial data streams, respectively. For the proximity sensor, we used shift registers (CD4021BE, Texas Instruments) that transform the parallel output from the *MCTs* to a single serial input to the microcontroller. The LEDs have an integrated control circuit to be connected in series. An additional printed circuit board connected to the main board contained 16 of the RGB-LEDs that receive the serial LED data stream from the microcontroller. We use this array of LEDs to convert the serial stream into parallel input to the *MCTs* by branching each input signal to the differential line drivers that transmit the signal to each *MCT*. To optimize the form factor of the interface unit, we made a third custom-printed circuit board that contains all connectors. In our current experiments, we assembled a circuit for connecting up to 16 *MCTs* but the *MaCaQuE* system would be easily expandable to a larger number. To set the transistors to drive the pumps of the reward systems, the 3.3V logic signal from the microcontroller is scaled up to 5V by a buffer (SN74HCT245N, Texas Instruments Inc, Dallas, Texas). As *MaCaQuE* incorporates parts operating at 3.3V (microcontroller), 5V (LED array), and 12V (peristaltic pump and *MCT*), we used a standard PC-power supply (ENP-7025B, Jou Jye Computer GmbH, Grevenbroich, Germany) as power source. Additionally, 12 digital general-purpose-input-output (GPIO) pins are available on the interface, which were used to 1) send and receive synchronizing signals to other behavioral or neural recording hardware (strobe); 2) add a button to manually control reward units; and 3) add a switch to select which reward unit is addressed by the manual reward control. Further options such as sending test signals or adding sensors or actuators are possible. Custom-printed circuit boards are designed with EAGLE version 6 (Autodesk Inc, San Rafael, California).

We used Arduino-C to program the microcontroller firmware. *MaCaQuE* was accessed by a USB connection from a computer using either Windows or Mac OS. A custom-written C++ software package (MoRoCo) operated the behavioral task and interfaced with *MaCaQuE* via the microcontroller. We developed hardware testing software using Processing and C++. *MaCaQuE* was also used in another study involving human participants (*Berger et al., 2019*). Schematics and software are available online (*Berger and Gail, 2020* and https://github.com/sensorimotorgroupdpz/MaCaQuE).

## Reach Cage

The Reach Cage is a cage-based training and testing environment for sensorimotor experiments with a physically unrestrained rhesus monkey (*Figure 1A*, *Video 1*). Inner cage dimensions are 170 cm x 80 cm x 85 cm (W x D x H) with a metal mesh grid at the top and bottom, a solid metal wall on one long side (back), and clear polycarbonate walls on all other sides. The idea of the experiment was to implement a memory-guided goal-directed reach task with instructed delay, equivalent to common conventional experiments (*Crammond and Kalaska, 2000*), to compare neural responses during planning and execution of reaches towards targets at different positions in space.

We used *MaCaQuE* to provide 10 visual cues and reach targets (*MCTs*) inside the cage (*Figure 1D*). Two *MCTs* were positioned on the floor pointing upwards. Eight were placed 25 cm below the ceiling in two rows of four each, pointing toward the middle position between the two *MCTs* on the floor. The floor *MCTs* provided the starting position for the behavioral task (start buttons). The monkey could comfortably rest its hands on the start buttons while sitting or standing in between. The row of ceiling *MCTs* closer to the starting position was placed with a 10 cm horizontal distance and 60 cm vertical distance to the starting position (near targets). We chose this configuration to provide a comfortable position for a rhesus monkey to reach from the starting positions to the near targets without the need to relocate its body. The second row of *MCTs* was positioned at

100 cm horizontal distance from the starting positions (far targets) requiring the animal to make steps towards the targets (*Figure 2B*). An 11[th] *MCT* was placed outside the cage in front of the monkey (in the starting position and facing the opposite wall) to provide an additional visual cue. For positive reinforcement training, *MaCaQuE's* reward systems can provide fluid reward through protected silicon and metal pipes into one of two small spoon-size stainless steel bowls mounted approx. 20 cm above the floor in the middle of either of the two long sides of the Reach Cage.

## Behavioral task

We trained both monkeys on a memory-guided walk-and-reach task with instructed delay (*Figure 2A*). When the *MCT* outside lit up, the monkeys were required to touch and hold both start buttons (hand fixation). After 400–800 ms, one randomly chosen reach target lit up for 400 ms indicating the future reach goal (cue). The animals had to remember the target position and wait for 400–2000 ms (memory period) until the light of the *MCT* outside changed its color to red without changing the luminance (go cue). The monkeys then had a 600 ms time window starting 200 ms after the go cue to release at least one hand from the start buttons. We introduced the 200 ms delay to discourage the animals from anticipating the go cue and triggering a reach prematurely. After releasing the start buttons, the animals needed to reach to the remembered target within 600 ms or walk-and-reach within 1200 ms dependent on whether the target was near or far. Provided the animals kept touching for 300 ms, the trial counted as correct indicated by a high pitch tone and reward. A lower tone indicated an incorrect trial. The reward was delivered as juice in one of two randomly assigned drinking bowls. We used unpredictable sides for reward delivery to prevent the animal from planning the movement to the reward before the end of the trial.

In the beginning, we did not impose the choice of hand on the monkeys in this study but let them freely pick their preferred hand. While monkey K reached to the targets with the right hand, monkey L used the left hand. Both animals consistently used their preferred hand and never switched. For the walk-and-reach task we trained monkey K to use its left hand using positive reinforcement training. Once trained, the monkey used consistently its left hand.

In a control session (*Figure 5—figure supplement 1*) we added a passage in the middle of the walk-and-reach movements. The session was split into two blocks with (160/100 trials for monkey K/L) and without (154/178 trials for monkey K/L) this passage. The passage had an opening of 31 cm horizontally that constrained the animal's walk-and-reach movements to a narrower path. Reach movements were unaffected.

All data presented in this manuscript were collected after animals were trained on the behavioral task.

## Motion capture and analysis of behavior

The animals' behavior was analyzed in terms of accuracy (percent correct trials), timing (as registered by the proximity sensors), and arm kinematics (video-based motion capture).

We analyzed start button release and movement times of both monkeys based on the *MCT* signals when they performed the walk-and-reach task (monkey K: 19 sessions; monkey L: 10 sessions). Button release time is the time between the go cue and the release of one of the start buttons. Movement time is the time between the release of one of the start buttons and target acquisition. We analyzed the timing separately for each monkey and separately for all near and all far targets.

Additionally, we tracked continuous head and arm kinematics in detail offline (). We recorded four video streams in parallel from different angles together with the *MCT* signals and the neural data. For these synchronized multi-camera video recordings, we used a commercial video capture system (Cineplex Behavioral Research System, Plexon Inc, Dallas, Texas) incorporating four Stingray F-033/C color cameras (Allied Vision Technologies GmbH, Stadtroda, Germany). Videos were recorded with 60 fps frame rate in VGA resolution. Video processing on camera and host PC takes less than 20 ms (camera shutter opening time not included). The system uses a central trigger to synchronize all cameras. For synchronization with all other data, the system sent a sync pulse every 90 frames to *MaCaQuE*.

To quantify the movement trajectories, we tracked the 3-dimensional position of the left wrist, elbow, shoulder, and headcap (part of the head implant, see below and *Figure 6C*, no 10) frame-by-frame when the monkeys performed the walk-and-reach task correctly. To do so, we first tracked the

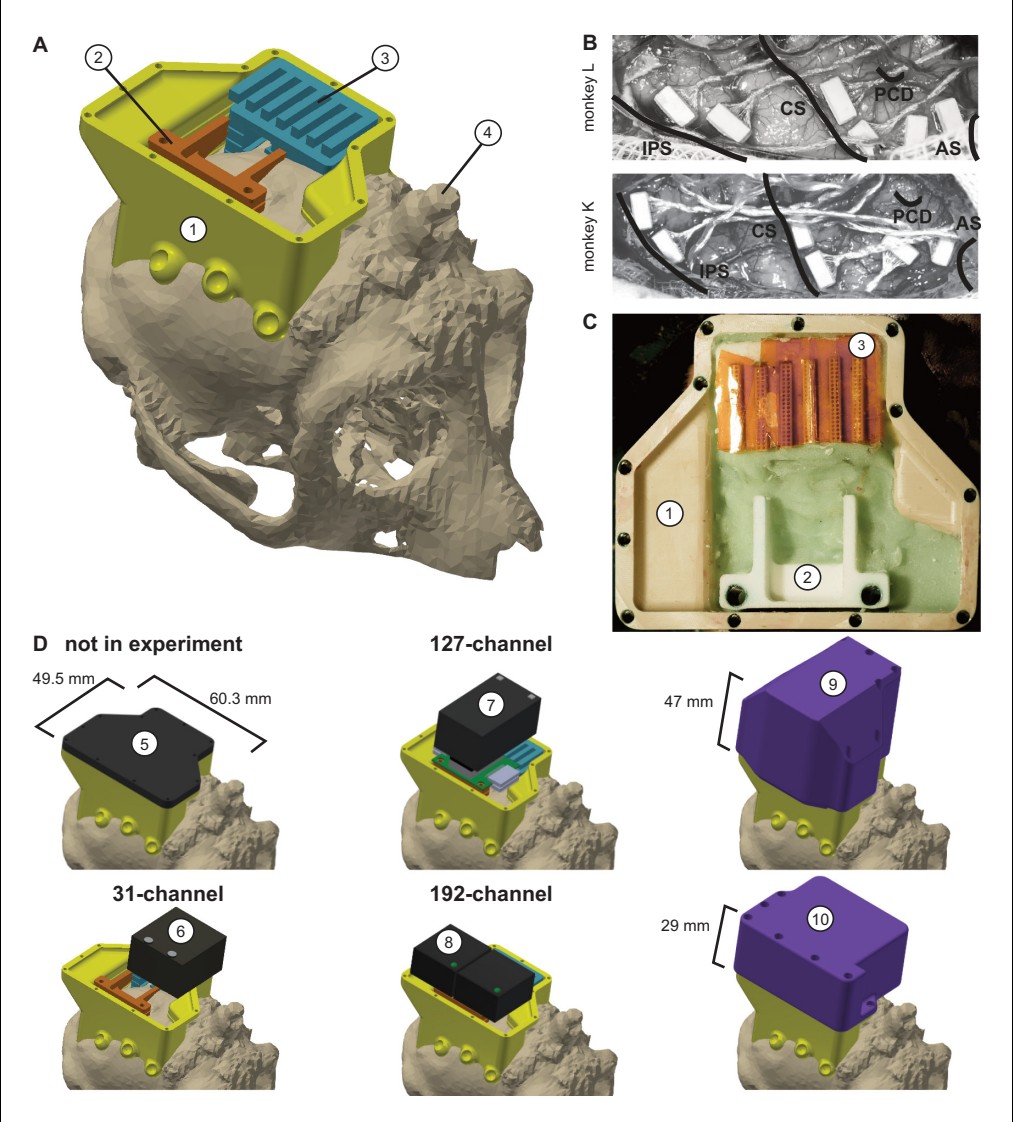

**Figure 6.** Implant system design. (**A**) Three-dimensional computer models of the implants and electronics. The skull model of monkey L (beige) is extracted from a CT scan including the titanium implant for head fixation (4, headpost), which was already implanted before this study. Further implants are colored for illustrative purposes and do not necessarily represent the actual colors. (**B**) Image of microelectrode array placement during the surgery of monkey L (top) and monkey K (bottom). Anatomical landmark descriptions: IPS – intraparietal sulcus; CS – central sulcus; PCD – postcentral dimple; AS – arcuate suclus. (**C**) Image of the implants on monkey L's head. (**D**) Different configurations of wireless headstages and protective headcaps temporally mounted on the implants. Numbers indicate: 1 – chamber; 2 – adapter board holder; 3 – array connector holder; 4 – headpost (from CT scan); 5 – flat headcap; 6 – W32 headstage; 7 – W128 headstage; 8 - Exilis headstage (two used in parallel); 9 – headcap for W128 headstage; 10 - headcap for W32 or Exilis headstages.

2-dimensional position in each video and then reconstructed the 3-dimensional position out of the 2-dimensional data. For 2-dimensional markerless body-part tracking we used DeepLabCut (DLC), based on supervised deep neural networks to track visual features consistently in different frames of a video (*Mathis et al., 2018*; *Mathis et al., 2019*; *Nath et al., 2019*). We trained a single network based on a 101-layer ResNet for all four cameras and both monkeys. Using DLC's own tools, we labeled in total 7507 frames from 12 sessions (four monkey K and eight monkey L). All training frames were randomly extracted from times at which the monkeys performed the walk-and-reach task correctly. We not only trained the model to track headcap, left wrist, elbow, and shoulder, but

also snout, left finger, right finger, wrist, elbow, shoulder, tail, and four additional points on the headcap. While those additional body parts were less often visible with this specific camera setting and not of interest for our current study, the tracking of certain desired features can be improved by training DLC models to additional other features (see *Mathis et al., 2018* for details). We used cross validation to estimate the accuracy of DLC in our situation, using 95% of our labeled data as training data for the model and 5% as test data. The model provides a likelihood estimate for each data point. We removed all results with a likelihood of less than 0.9. For the remaining data points of all 10 features, the root mean squared error was 2.57 pixels for the training and 4.7 pixels for test data. With this model we estimated the position of the body parts in each video. Then we reconstructed the 3-dimensional position using the toolbox pose3d (*Sheshadri et al., 2020*). First, we captured images from a checkerboard with defined length on all four cameras at the same time. Using the Computer Vision Toolbox from Matlab (Mathworks Inc, Natick, Massachusetts), we estimated the camera calibration parameters for each camera and for each camera pair. Pose3d uses those parameters to triangulate the 3-dimensional position from at least two camera angles. If feature positions from more than two cameras are available, pose3d will provide the least-squares estimate. By projecting the 3-dimensional position back into 2-dimensional camera coordinates we could measure the reprojection error. We excluded extreme outliers with a reprojection error above 50 pixels for at least one camera.

After the reconstruction of the 3-dimensional positions of the body parts, we performed an outlier analysis. First, we applied a boundary box with the size of 132 cm x 74 cm x 75 cm (W x D x H) and removed data points that lay outside the box. Second, we looked for outliers based on discontinuity over time (aka speed). We calculated the Euclidean distance between each consecutive time point for each body part trajectory and applied a threshold to detect outliers. We only rejected the first and every second outlier as a single outlier will lead to two 'jumps' in the data. Then we reiterated the process until all data points were below threshold. We applied different thresholds for each body part, dependent on whether the frame was during a movement (between start button release and target acquisition) or not. Specifically, we used 12 mm/frame and 80 mm/frame for the wrist and 15 mm/frame and 40 mm/frame for the other body parts with the higher threshold during the movement. With a frame rate of 60 fps, 100 mm/frame corresponds to 6 m/s. After rejecting all outliers (DeepLabCut low likelihood, reprojection error, boundary box, and discontinuity), the percentage of valid data points of all seven analyzed sessions during correctly performed trials for Monkey K/L was: wrist 94.93%; elbow 92.51%; shoulder 94.98%; headcap 97.58%. We interpolated the missing data points using phase preserving cubic interpolation.

We analyzed the movement trajectories of the four body parts during reach and walk-and-reach movements. For the behavioral analysis (2/3 sessions, 469/872 successful trials monkey K/L), we chose a time window of between 100 ms before start button release and 100 ms after target acquisition (*Figure 3*). For the analysis with neural data (231/326 successful trials monkey K/L one session each), we chose the time window of between 300 ms before start button release and 300 ms after target acquisition (*Figure 5*). In both cases, we used linear interpolation for temporal alignment of the data between trials and relative to the neural data in the latter case. For trial-averaging, we averaged the data across trials on each aligned time point for each dimension. The 3-dimensional data are presented from a side-view (*Figure 3*) and top-view (*Figure 5A*, *Figure 5—figure supplement 1A*) of the movement. The side-view is defined by one of the four cameras directly facing the side of the Reach Cage. Arm posture plots are straight lines connecting wrist with elbow, elbow with shoulder, and shoulder with headcap. For the variability analysis, we calculated the Euclidean distance at each time point and trial to the trial-averaged trajectory for each target and body part. We then averaged the distances over all time points for each trial and present the median and 0.75-quartile for each body part and target distance pooled over the target position. For the control session with a narrow passage (*Figure 5—figure supplement 1A*, 314/278 successful trials monkey K/L one session each), we additionally analyzed the spread of the wrist and head position of the walk-and-reach movements over trials at a 40 cm distance from the animals' average wrist starting position. We report range and s.d. over the axis orthogonal to the side-view, that is the target axis, and used the Kolmogorow-Smirnow test to determine whether distributions with and without narrow passage differed.

The behavioral analyses were performed using Matlab with the data visualization toolbox *gramm* (*Morel, 2018*). The 2-dimensional feature tracking with DeepLabCut was done in Python (Python Software Foundation, Beaverton, Oregon).

## Implant system design

Wireless neural recordings from the cerebral cortex of rhesus monkeys during minimally restrained movements require protection of the electrode array connectors and the headstage electronics of the wireless transmitters. We designed a protective multi-component implant system to be mounted on the animal skull (*Figure 6*). The implant system and implantation technique was designed to fulfill the following criteria: 1) Electrode connectors need to be protected against dirt and moisture; 2) While the animal is not in the experiment, the implants need to be small and robust enough for the animal to live unsupervised with a social group in an enriched large housing environment; 3) During the experiment, the wireless headstage needs to be protected against manipulation by the animal and potential physical impacts from bumping the head; 4) The head-mounted construction should be as lightweight as possible; 5) Placing of the electrode arrays and their connectors during the surgery needs to be possible without the risk of damaging electrodes, cables, or the brain; 6) Implant components in contact with organic tissue need to be biocompatible; 7) Temporary fixation of the animal's head in a primate chair needs to be possible for access to implants and for wound margin cleaning; 8) Implants must not interfere with wireless signal transmission; 9) Optionally, the implant may serve as trackable object for motion capture.

We designed the implant system for two main configurations: first, a home configuration containing only permanently implanted components and being as small as possible when the animal is not in a recording session but in its group housing (*Figure 6D*, top left); second, a recording configuration with removable electronic components being attached. This configuration should fit a 31-channel headstage, a 127-channel headstage (W32/W128, Triangle BioSystems International, Durham, North Carolina), or two 96-channel headstages (CerePlex Exilis, Blackrock Microsystems LLC, Salt Lake City, Utah). Headstage placement is illustrated in *Figure 6D*. The implant system consists of four custom-designed components: a skull-mounted outer encapsulation (chamber; *Figure 6A/C*, no 1), a mounting base for holding a custom-designed printed circuit board (adaptor board holder, no 2), a mounting grid to hold the connectors of the electrode arrays (connector holder, no 3), and a set of different-sized caps to contain (or not) the different wireless headstages (no 5–10). Dimensions of the wireless headstages are W32: 17.9 mm x 25 mm x 14.2 mm (W x D x H), 4.5 g weight; W128: 28.7 mm x 34.3 mm x 14.2 mm (W x D x H), 10 g weight; Exilis: 25 mm x 23 mm x 14 mm (W x D x H), 9.87 g weight.

We designed the implants custom-fit to the skull using CT and MRI scans. Using 3D Slicer (Brigham and Women's Hospital Inc, Boston, Massachusetts), we generated a skull model out of the CT scan (*Figure 6A*) and a brain model out of the MRI scan (T1-weighted; data not shown). In the MRI data we identified the target areas for array implantation based on anatomical landmarks (intraparietal, central, and arcuate sulci; pre-central dimple), and defined Horsley-Clarke stereotactic coordinates for the craniotomy necessary for array implantation (*Figure 6B*). We used stereotactic coordinates extracted from the MRI scan to mark the planned craniotomy on the skull model from the CT scan. We then extracted the mesh information of the models and used Inventor (Autodesk Inc, San Rafael, California) and CATIA (Dassault Systèmes, Vélizy-Villacoublay, France) to design virtual 3-dimensional models of the implant components which are specific to the skull geometry and planned craniotomy. Both monkeys already had a titanium headpost implanted, of which the geometry, including subdural legs, was visible in the CT (*Figure 6A*, no 4), and, therefore, could be incorporated into our implant design.

We built the chamber to surround the planned craniotomy and array connectors (*Figure 6A/C*, no 1). The chamber was milled out of polyether ether ketone (TECAPEEK, Ensinger GmbH, Nufringen, Germany) to be lightweight (monkey K/L: 10/14 grams; 65/60.3 mm max. length, 50/49.5 mm max. width, 24.9/31.2 mm max. height; wall thickness: 2/2 mm) and biocompatible. For maximal stability despite low diameter, stainless-steel M2 threads (Helicoil, Böllhoff, Bielefeld, Germany) were inserted into the wall for screwing different protective headcaps onto the chamber. The built-in eyelets at the outside bottom of the chamber wall allow mounting of the chamber to the skull using titanium bone screws (2.7 mm corticalis screws, 6–10 mm length depending on bone thickness, DePuy

Synthes, Raynham, Massachusetts). Fluting of the lower half of the inner chamber walls let dental cement adhere to the chamber wall.

The subdural 32-channel floating microelectrode arrays (FMA, Microprobes for Life Science) are connected by a stranded gold wire to an extra-corporal 36-pin nano-strip connector (Omnetics Connector Corporation, Minneapolis, Minnesota). We constructed an array connector holder to hold up to six of the Omnetics connectors inside the chamber (*Figure 6A/C*, no 3). The connector holder was 3D-printed in a very lightweight but durable and RF-invisible material (PA2200 material, Shapeways). The holding grid of the array connector holder is designed such that it keeps the six connectors aligned in parallel with 2 mm space between. The spacing allows for: 1) connecting six 32-channel Cereplex (Blackrock Microsystems LLC) headstages for tethered recording simultaneously on all connectors, 2) directly plugging a 31-channel wireless system onto one of the array connectors, or 3) flexibly connecting four out of six arrays with adaptor cables to an adaptor board, linking the arrays to a 127-channel wireless system. The total size of the array connector is 27 mm x 16.2 mm incorporating all six connectors. The bottom of the array connector holder fits the skull geometry with a cut-out to be placed above an anchor screw in the skull for fixation with bone cement (PALACOS, Heraeus Medical GmbH, Hanau, Germany). This is needed as the array connector is placed on the skull next to the craniotomy during insertion of the electrode arrays, that is before implantation of the surrounding chamber (see below). The medial side of the holding grid, pointing to the craniotomy, is open so that we can slide in the array connectors from the side during the surgery. On the lateral side small holes are used to inject dental cement with a syringe to embed and glue the connectors to the grid.

The 31-channel wireless headstage can be directly plugged into a single Omnetics nano-strip array connector. The 127-channel wireless headstage instead has Millmax strip connectors (MILL-MAX MFG. CORP., Oyster Bay, New York) as input. A small adapter board (electrical interface board, Triangle BioSystems International) builds the interface to receive up to four Omnetics nano-strip connectors from the implanted arrays via adaptor cables (Omnetics Connector Corporation). We constructed a small holder with two M3 Helicoils for permanent implantation to later screw-mount the adaptor board when needed during recording (*Figure 6A/C*, no 2). Fluting on the sides of the adaptor board holder helps embedding of the holder into dental cement. Like the array connector holder, the adaptor board holder was 3D-printed in PA2200. The 96-channel Exilis headstages have three Omnetics nano-strip connectors which would fit into the array connectors; however, precise alignment was very difficult because of the small size of the connector. Instead we relied on adapter cables, as with the 127-channel headstage, to connect headstage and array connectors. The two headstages fit perfectly in the protective headcap (*Figure 6D*, no 10), which also prevents movements of the headstage itself.

Depending on the experiment and space needed, we used three different protective headcaps. While the animal was not in an experiment, a flat 4 mm machine-milled transparent polycarbonate headcap with rubber sealing protected the connectors against moisture, dirt, and manipulations (*Figure 6D*, no 5). During experiments, we used two specifically designed protective headcaps for the two different wireless headstages. Both were 3D-printed in PA2200 in violet color to aid motion capture. As the 31-channel wireless headstage is connected to the array connectors directly, it extends over the chamber walls when connected to one of the outermost connectors (*Figure 6D*, no 6). We designed the respective protective headcap to cover this overlap (*Figure 6D*, no 10). The 127-channel wireless headstage (*Figure 6D*, no 7) with its adapter board is higher and overlaps the chamber on the side opposite to the connectors. We designed the respective headcap accordingly (*Figure 6D*, no 9). The two 96-channel Exilis Headstages were used with the smaller headcap (no 10). For Monkey L, we 3D-printed a version with slightly larger inner dimensions in green polylactic acid (PLA) using fused deposit modeling.

As the 3D-printed headcaps were only used during recording sessions, that is for less than 2 h, without contact to other animals, and under human observation, we did not add extra sealing against moisture. However, by adding rubber sealing, the internal electronics would be safe even for longer periods of time in a larger and enriched social-housing environment without human supervision.

## Surgical procedure

The intracortical electrode arrays and the permanent components of the chamber system were implanted during a single sterile surgery under deep gas anesthesia and analgesia via an IV catheter. Additionally, the animals were prophylactically treated with phenytoin (5–10 mg/kg) for seizure prevention, starting from 1 week before surgery and continuing until 2 weeks post-surgery (fading-in over 1 week), and with systemic antibiotics (monkey K: cobactan 0.032 ml/kg and synolux 0.05 ml/kg 1 day pre-surgery and 2 days post-surgery; monkey L: duphamox, 0.13 ml/kg, 1 day pre-surgery to 1 day post-surgery). During craniotomy, brain pressure was regulated with mannitol (monkey K/L: 16/15.58 ml/kg; on demand). Analgesia was refreshed on a 5 h cycle continuously for 4 post-surgical days using levomethadon (0.28/0.26 mg/kg), daily for 1/3post-surgical days using metacam (0.24/ 0.26 mg/kg) and for another 4 days (rimadyl, 2.4/1.94 mg/kg) according to demand.

We implanted six FMAs in the right hemisphere of both monkeys. Each FMA consists of 32 parylene-coated platinum/iridium electrodes and four ground electrodes arranged in four rows of nine electrodes (covering an area of 1.8 mm x 4 mm) staggered in length row-wise, with the longest row opposite the cable and the shortest row closest to the cable. Two FMAs were placed in each of the three target areas: parietal reach region (PRR), dorsal premotor cortex (PMd), and arm-area of primary motor cortex (M1). MRI scans were used to define desired array positions and craniotomy coordinates. As we did not know the location of blood vessels beforehand, the final placing of the arrays was done based on the visible anatomical landmarks. PRR arrays were positioned along the medial wall of the intraparietal sulcus (IPS) starting about 7 mm away from the parieto-occipital sulcus (*Figure 6B*), with electrode lengths of 1.5–7.1 mm. M1 arrays were positioned along the frontal wall of the central sulcus, at a laterality between precentral dimple and arcuate spur, with electrode lengths of 1.5–7.1 mm. The longer electrodes of PRR and M1 arrays were located on the side facing the sulcus. PMd arrays were positioned, between arcuate spur, precentral dimple, and the M1 arrays as close to the arcuate spur, with electrode lengths of 1.9–4.5 mm.

Except for the steps related to our novel chamber system, the procedures for FMA implantation were equivalent to those described in *Schaffelhofer et al., 2015*. The animal was placed in a stereotaxic instrument to stabilize the head and provide a Horsley-Clarke coordinate system. We removed skin and muscles from the top of the skull as much as needed based on our pre-surgical craniotomy planning. Before the craniotomy, we fixed the array connector holder to the skull with a bone screw serving as anchor and embedded in dental cement on the hemisphere opposite to the craniotomy. After removing the bone with a craniotome (DePuy Synthes) and opening the dura in a U-shaped flap for later re-suturing, we oriented and lowered the microelectrode arrays one-by-one using a manual micro-drive (Narishige International Limited, London, UK), which was mounted to the stereotaxic instrument on a ball-and-socket joint. Before insertion, the array connector was put into our array connector holder and fixed with a small amount of dental cement. During insertion, the array itself was held at its back plate under-pressure in a rubber-coated tube connected to a vacuum pump which was attached to the microdrive. We slowly lowered the electrodes about 1 mm every 30 s until the back plate touched the dura mater. We let the array rest for 4 min before removing first the vacuum and then the tube.

After implanting all arrays, we arranged the cables for minimal strain and closed the dura with sutures between the cables. We placed Duraform (DePuy Synthes) on top, returned the leftover bone from the craniotomy and filled the gaps with bone replacement material (BoneSource, Stryker, Kalamazoo, Michigan). We sealed the craniotomy and covered the exposed bone surface over the full area of the later chamber with Super-Bond (Sun Medical Co Ltd, Moriyama, Japan). We secured the array cables at the entry point to the connectors and filled all cavities in the array connector holder with dental cement. We mounted the chamber with bone screws surrounding implants and craniotomy, positioned the adaptor board holder, and filled the inside of the chamber with dental cement (*Figure 6C*). Finally, we added the flat protective headcap on the chamber.

## Neural recordings

Neural recordings were conducted in both monkeys during the walk-and-reach task in the Reach Cage. We recorded wirelessly from all six arrays simultaneously using the two 96-channel Exilis Headstages. To remove interference between the two headstages, we placed a small metal plate between the two headstages which was connected to the ground of one headstage. We used seven

antennas in the cage, which were all connected to both receivers for the respective headstage. The headstages used carrier frequencies of 3.17 GHz and 3.5 GHz, respectively. The signal was digitized on the headstages and sent to two recordings systems, one for each headstage. We used a 128-channel Cerebus system and a 96-channel CerePlex Direct system (both Blackrock Microsystems LLC) for signal processing. For the control session (*Figure 5—figure supplement 1*), we sued a single 256-channel CerePlex Direct system (Blackrock Microsystems LLC) receiving input from both receivers. *MaCaQuE* sent the trial number at the beginning of each trial to the parallel port of each system. We connected an additional shift register M74HC595 (STMicroelectronics) to the GPIO port of MaCaQuE for interfacing the parallel ports. The recording systems recorded the trial number along with a time stamp for offline data synchronization.

We calculated data loss rate per trial on the broadband data. The headstage transmits digital data. When it loses connection the recording system repeats the latest value. As wireless data are transmitted in series, a connection loss affects all channels. We looked in the first 32 channels of the broadband data at least four consecutive times for which the data did not change. Then we labeled all consecutive time points as 'data lost' for which the data did not change. We did this for both 96-channel recordings separately. As we wanted to estimate the reliability of the 192-channel recording, we considered data loss at times where even only one of the two headstages showed data loss. Then we calculated the percentage of time points with data loss for each session only considering times within trials for which the monkey performed the task correctly. We also calculated the data loss for each trial separately. Only trials with data loss smaller than 5% were considered for further analysis.

We performed the preprocessing of broadband data and the extraction of waveforms as previously described (*Dann et al., 2016*). First, the raw signal was high-pass filtered using a sliding window median with a window length of 91 samples (~3 ms). Then, we applied a 5000 Hz low-pass using a zero-phase second order Butterworth filter. To remove common noise, we transformed the signal in PCA space per array, removed principle components that represented common signals, and transformed it back (*Musial et al., 2002*). On the resulting signal, spikes were extracted by threshold crossing using a negative or positive threshold. We sorted the extracted spikes manually using Offline Sorter V3 (Plexon Inc, Dallas, Texas). If single-unit isolation was not possible, we assigned the non-differentiable cluster as multi-unit, but otherwise treated the unit the same way in our analysis. We performed offline sorting for the example units (*Figure 4*), decoding and encoding analysis (*Figure 5B*) but not for the control session (*Figure 5—figure supplement 1B*). The spike density function for the example units (*Figure 4*) wereas computed by convolving spike trains per trial and per unit with a normalized Gaussian with standard deviation of 50 ms. The spike density function was sampled at 200 Hz. The exemplary broadband data in *Figure 4* show the data before preprocessing.

We analyzed the firing rate of all 192 channels in the 12 sessions and of four example units with respect to four different temporal alignments: target cue onset, go cue, start button release, and target acquisition. To quantify neural activity during the delay period and the movement, we analyzed time windows of 500 ms either immediately before or after a respective alignment. We analyzed the modulation of firing rate relative to the position of the reach targets and time window for each unit. We calculated an ANOVA with factors: distance (near, far), position (outer left, mid left, mid right, outer right), and time (before and after the respective alignments, eight time windows). We considered a channel/unit task modulated if there was a significant effect on any factor or interaction. We considered it position modulated if there was a significant main effect on position or an effect on position x distance, position x time, or position x distance x time.

For the population decoding analysis, we used a linear support vector machine (SVM) on the firing rate within 300 ms time windows. We decoded left vs. right side, that is grouped left-outer and left-mid targets as well as right-outer and right-mid targets. Reach and walk-and-reach movements were analyzed separately. Decoding accuracy was estimated by 20-fold cross validation. The 20-folds always referred to the same trials in each window throughout the timeline. For statistical testing we focused on one time window during memory and one during movement period, respectively. As the shortest trials have a memory period of 400 ms we selected 100–400 ms after the cue as the window for the memory period. For the movement period, we selected 300–0 ms before target acquisition. To determine decoding accuracy above chance level, we performed a one-tailed permutation test as follows. We generated a null distribution of 500 samples by permuting the target direction (left vs.

right category) in all trials independently for each monkey and distance, and calculated the 20-fold cross validated decoding accuracy as described before.

For the population encoding analysis, we calculated the population average of the firing rate modulation individually for each monkey, distance, and movement period as follows: First, we calculated the average firing rate for each unit and target direction; then we took the absolute difference between left and right target directions (same grouping as in the decoding analysis) and averaged the absolute difference over all units per area. We performed a one-tailed permutation test to test if this firing rate difference is significantly higher than expected by chance. For this, we generated a null distribution of 1000 samples and then calculated the population average of the firing rate modulation for each sample. For both permutation tests, a p-value was calculated by the fraction of the null distribution above the value to test. We used Bonferroni multiple comparison correction with a multiplier of 12 (3 areas x 2 distance x two time periods).

For the control session with and without a narrow passage for walk-and-reach movements (*Figure 5—figure supplement 1B*), we performed an SVM decoding analysis and calculated the population averaged firing rate modulation. We used 10-fold cross validation and tested if the decoding accuracy changed depending on whether or not the passage was present. To test if the passage had an effect, we used a two-tailed permutation test with 500 surrogate samples by permuting the passage label in all trials independently for each monkey. For the population encoding analysis, we calculated the firing rate modulation by target direction per channel, subtracted the modulation without passage from the modulation with passage, and calculated the average over the population per area. To test if the passage changed the left vs. right firing rate modulation, we generated a null distribution of 1000 samples and calculated the difference in firing rate modulation for each sample. For the permutation tests of the control session, we considered both tails of the null distribution to calculate the p-value. We applied Bonferroni multiple comparison correction with a multiplier of 6 (3 areas x two time periods).

Raw data and spike data processing was performed with Matlab and visualized using the toolbox *gramm* (*Morel, 2018*).

## Acknowledgements

We thank Sina Plümer for help with data collection and technical support, Klaus Heisig and Marvin Kulp for help with mechanical constructions, Swathi Sheshadri, Benjamin Dann, Mariana Eggert Martínez, and Baltasar Rüchardt for help with motion capture, Peer Strogies for help with implant design, Attila Trunk and Ole Fortmann for help with data collection, Pierre Morel, Enrico Ferrea, Michael Fauth, Jan-Matthias Braun, Christian Tetzlaff, and Florentin Wörgötter for helpful discussions, Leonore Burchardt for help with animal training, and Janine Kuntze, Luisa Klotz, and Dirk Prüße for technical support.

## Additional information

### Funding

| Funder | Grant reference number | Author |
|--------|------------------------|--------|
| Deutsche Forschungsgemeinschaft | DFG RU-1847-C1 | Alexander Gail |
| European Commission | EC-H2020-FETPROACT-16 732266 WP1 | Alexander Gail |

The funders had no role in study design, data collection and interpretation, or the decision to submit the work for publication.

### Author contributions

Michael Berger, Conceptualization, Data curation, Software, Formal analysis, Validation, Investigation, Visualization, Methodology; Naubahar Shahryar Agha, Investigation, Methodology; Alexander Gail, Conceptualization, Resources, Supervision, Funding acquisition, Methodology, Project administration

## Author ORCIDs
Michael Berger (iD) https://orcid.org/0000-0002-7239-1675
Alexander Gail (iD) https://orcid.org/0000-0002-1165-4646

## Ethics

Animal experimentation: Both animals were housed in social groups with one (monkey L) or two (monkey K) male conspecifics in facilities of the German Primate Center. The facilities provide cage sizes exceeding the requirements by German and European regulations, access to an enriched environment including wooden structures and various toys (Calapai et al. 2017). All procedures have been approved by the responsible regional government office [Niedersächsisches Landesamt für Verbraucherschutz und Lebensmittelsicherheit (LAVES)] under permit numbers 3392 42502-04-13/1100 and comply with German Law and the European Directive 2010/63/EU regulating use of animals in research.

## Decision letter and Author response
Decision letter https://doi.org/10.7554/eLife.51322.sa1
Author response https://doi.org/10.7554/eLife.51322.sa2

# Additional files

## Supplementary files
• Supplementary file 1. Overview of neurophysiology studies with unrestrained monkeys. This table presents an overview of current neurophysiology studies with unrestrained monkeys. The Reach Cage provides the only environment capable of instructing the animal to control start and end times of a desired movement, which for example allows training of animals to withhold a movement and study movement planning. Also, while previous studies studied a variety of behavior, instructed goal-directed movements were always direct food (source) directed movements. Only the Reach Cage can dissociate motor goals from food sources. There are four other studies that present multiple movement goals. There are locomotion studies that incorporate 3D motion capture, but these are not markerless and none showed 3D kinematics of reaching behavior. Note that other studies have shown 3D markerless motion capture of freely behaving monkeys (*Bala et al., 2020*; *Nakamura et al., 2016*); however, without neurophysiological recordings.

• Transparent reporting form

## Data availability
All data (schematics, soft- and hardware documentation) for constructing the MaCaQuE or equivalent systems is made available via GitHub: https://github.com/sensorimotorgroupdpz/MaCaQuE.

The following dataset was generated:

| Author(s) | Year | Dataset title | Dataset URL | Database and Identifier |
|---|---|---|---|---|
| Berger M, Gail A | 2020 | sensorimotorgroupdpz/MaCaQuE | https://doi.org/10.5281/zenodo.3685793 | Zenodo, 10.5281/zenodo.3685793 |

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
