## [Decision Letter]

**Acceptance summary:**

In this manuscript, the authors present an experimental environment and approach to instruct and quantify behavior in unrestrained macaque monkeys while conducting high-bandwidth wireless recordings from multiple brain areas simultaneously. The results demonstrate how the novel approach and methodology can be used to resolve neural mechanisms of behavior in more naturalistic settings.

**Decision letter after peer review:**

[Editors’ note: the authors submitted for reconsideration following the decision after peer review. What follows is the decision letter after the first round of review.]

Thank you for submitting your work entitled "The Reach Cage environment for wireless neurophysiology during structured goal-directed behavior of unrestrained monkeys" for consideration by *eLife*. Your article has been reviewed by three peer reviewers, and the evaluation has been overseen by a Reviewing Editor and a Senior Editor. The following individuals involved in review of your submission have agreed to reveal their identity: Samantha Santacruz (Reviewer #2); Andrew Jackson (Reviewer #3).

Our decision has been reached after consultation between the reviewers. Based on these discussions and the individual reviews below, we regret to inform you that your work will not be considered further for publication in *eLife* at this time. Our assessment is that the revisions required may require a substantial reworking of the study to overcome what might be significant technical limitations. However, the reviewers found enough merit in this study that we would be happy to consider a new submission if you are able to adequately address the concerns raised.

In particular, the reviewers found several aspects of the work of value, such as the timeliness, the design and use of behavioral cues in the cage, and the combination of remote neurophysiological and behavioral tracking. This led to broad support for the goals and methods of the work. However, all three reviewers highlighted major weaknesses in the methodological advance presented that substantially undercut the significance of the work. The fundamental concern is that the behavioral apparatus alone seems more suited to a more specialized methods journal and it is not clear what kind of important scientific advance the wireless and motion capture components can support.

More clarification and quantification of the motion tracking performance is needed. The successful frame capture rate needs to be quantified over intervals of fixed duration and how this compares with reported performance of similar motion tracking methods that are not in-cage needs to be presented. Data for the second subject, Monkey L, also needs to be presented to give a second animal that would strengthen the claims of overall system performance. The inability to monitor more than 1 DOF is also considered a major limitation for a system meant ultimately to examine modulation of activity in an unconstrained animal, and this needs to be addressed.

The wireless recordings also need to be analyzed to quantify in detail wireless performance for the 31 and 127 channel recording setups using standard metrics, such as BER. Since the work is done in a non-ideal, highly reflective environment, we would be satisfied if the performance was close to the worst-case reported BER value of 10^-2 (taken from Yin et al., 2014, from which the current equipment is derived).

More detailed comments from the individual reviewers are provided below.

Reviewer #1:

This manuscript represents behavioral paradigms and data collection methods that are certain to become the focus of attention in the next few years. The study of the relations between single-neuron activity and highly constrained motor behaviors has largely given way in the past ten years to the study of large numbers of simultaneously recorded neurons, albeit, with largely the same behavioral paradigms that have been in place for the past several decades. The advent of the study of more natural behaviors has only more recently begun, given the attendant difficulties of designing useful behavioral paradigms, monitoring the movements themselves, and recording large number of neurons wirelessly.

This study is an initial attempt in that direction, and is described frequently as a proof-of-concept. The MaCaQuE (Macaque Cage Query Extension) component, in addition to its clever name, is the most original and important part of the methods. It combines proximity touch sensors and multi-color cue LEDs with a well-designed network communication method that interconnects the remote components with a microcontroller. The system is expandable and flexible, and forms the basis of a nice behavioral control system for eliciting innovative, more natural behaviors. I also found the flexible head-mounted chamber and connector-mounting system for neural recordings to be innovative and potentially quite useful. It is designed for an Omnetics-based headstage, but could presumably be modified without too great effort for the Cerebus connector.

Unfortunately, the cage itself appears to have been designed with too much metal for good RF transmission. While the 32-channel transmitter functioned adequately, the 128-channel version did not, suffering unacceptable artifacts and signal loss. While 32 channels will certainly be useful for many studies, it is a good bit less than the ~100 channels that have become something of the norm for multi-channel studies these days. Beyond that, passing reference to illumination levels and perhaps other restrictions seem to have severely limited motion tracking. Although they used four cameras to track a single colored marker (dye on the fur) at the monkey's wrist), the percentage of tracked frames seems to have been very low. Although this too, was described in less detail than would have been ideal, apparently one-third of trials tracked fewer than ~150ms of the trajectory ("5 data points"). These limitations of data collection strike me as being pretty substantial.

In addition to the methodological considerations reported here, this study also includes basic results for recordings from three cortical motor areas during "stretch and reach" and "walk and reach" paradigms. These results are presented in sufficiently limited form that they serve primarily to illustrate the paradigms themselves. I am sure that a fuller-length treatment of these data is planned, and anticipate that it will be of great interest. Beyond the behavioral and data collection methods described here, these kinds of experiments will also require a range of more sophisticated analytical approaches to take full advantage of the data.

Consequently, despite its potential, this paper is not adequately compelling either in term of methodological or scientific contributions. Addressing the former will be difficult, unless I have misunderstood the basic limitations of the wireless neural recordings and motion tracking. I assume the latter is a distinct possibility. There are a fair number of grammatical errors that should be addressed, not all of which I attempted to mark.

Reviewer #2:

The authors present a behavior system (the Reach Cage in combination with MaCaQuE interaction device) for performing sensorimotor tasks with nonhuman primates while wirelessly recording neural data. Overall, I find the system design to be novel and a clear demonstration of a system that may be used for the suggested purposes. My two main concerns are: (1) the motivation for the need of such a system, and (2) the limitation of kinematic tracking to one position (wrist). Regarding the former concern, I think the authors could improve their introductory discussion on how such a system is lacking and provides greater insight than, for example, using an existing commercial wireless recording system (e.g. from Blackrock) with something as simple as a cage-mounted touchscreen (essentially porting standard experimental setups with head-fixed animals to a the homecage environment). Regarding the latter concern, the authors do address this point late in the paper (Discussion section) but unconvincingly. There are sleeves/bodysuits that could be used with LED markers for tracking multiple positions that subjects do tolerate. The authors should at least make an argument regarding potential ways they could envision future revisions of their system to allow for tracking of more than one point. Additionally, I have minor concerns regarding the statistical analysis of single-unit data (present in Figure 4). For the analysis of Example B, statistical tests are performed separately for the two time points (cue onset and before go cue) and for the two distances of targets (near and far). It is more appropriate to do a combined statistical analysis, such as a two-way ANOVA with firing rate as the dependent variable, and timepoint (2 levels) and distances (2 levels) are independent variables. Similarly, for example D, a combined statistical analysis should be performed for the distances (near and far) and sides (left and right).

Reviewer #3:

This methods paper by Berger and Gail describes an experimental set-up for studying goal-directed reaching in unrestrained non-human primates with associated wireless neural recording. This work is timely, since the use of chronically-implanted electrodes and wireless recording is allowing neuroscientific experiments to be conducted in less constrained environments. However, the lack of constraints brings both advantages (behaviours may be more naturalistic than conventional restrained tasks such as centre-out reaching) and disadvantages (behaviours are less controlled and therefore data is less amenable to common analysis techniques such as trial averaging). The approach here appears to be a reasonable trade-off between the two, in that movements are unrestrained but trained/cued so as to be fairly stereotyped. I can certainly envisage some scientific questions that would be amenable to such an approach, although with the caveat that such stereotyped behaviours may not be only marginally more naturalistic than the traditional tasks. I also see this work as valuable as a stepping stone to studying fully unconstrained, free behaviour, although in this regard it may be important to incorporate more detailed motion capture in order to make sense of the recordings.

The neural recording uses a commercially available system, and only a couple of exemplar neurons are reported, so the principle novelty of this work comprises a description of the behavioural set-up, task design and kinematic recordings. The system comprises a number of cue/target boxes placed within an arena, with camera-based tracking of movement, and the operation of this is well described in the paper. However, it would be useful to include more description of how easily the animals were trained on this task (e.g. how many sessions, how long did it take for consistent behaviour to emerge). The authors should also assess how stereotyped was the behaviour from trial-to-trial vs. session-to-session. The collection of motion kinematics seems to be fairly limited – only the position of the wrist is tracked via red paint markings, in only one of the animals. Why was this not done for the other animal? No attempt is made to characterise the accuracy of this tracking – it appears that quite a large region of the wrist was painted so I can envisage that determining the centre location from multiple cameras may be quite dependent on arm/hand orientation etc. It would also be interesting to discuss whether other colours on, for example, the elbow and shoulder could allow a more detailed reconstruction of limb kinematics.

The modular design of the headpiece is nice. However, please add a scale bar to Figure 5C and also give dimensions for the total implant height. It is not clear whether the wireless recording system is battery- or inductively-powered. If the former, please give details of battery capacity, life-time, weight etc. If the latter, please include details of how power coils were positioned etc.

[Editors’ note: further revisions were suggested prior to acceptance, as described below.]

Thank you for submitting your article "The Reach Cage environment for wireless neural recordings during goal-directed behavior of unrestrained monkeys" for consideration by *eLife*. Your article has been reviewed by three peer reviewers, and the evaluation has been overseen by a Reviewing Editor and Joshua Gold as the Senior Editor. The following individual involved in review of your submission has agreed to reveal their identity: Samantha R Santacruz (Reviewer #1).

The reviewers have discussed the reviews with one another and the Reviewing Editor has drafted this decision to help you prepare a revised submission.

Summary:

The authors present an updated version of their behavior system (the Reach Cage in combination with MaCaQuE) which has been demonstrated in a sensorimotor task ("walk-and-reach" task) with nonhuman primates while wirelessly recording neural data as a proof-of-principle of the overall functionality of the system. The primary strength of this work, is perceived by the reviewers to be the reduced constraints on behavior (towards naturalistic settings) while still affording systematic study via controlled tasks and quantification. Previous concerns regarding the motivation for the study and limitations of kinematic tracking have been addressed with a marked significant improvement in the system. However, since the individual pieces of the system have been previously demonstrated (e.g. wireless recordings, in-cage behavior), whether the combined effort represents a novel contribution in a way that will facilitate new experiment needs more direct support. Moreover, more detailed analyses of the behavioral and neurophysiological data are needed.

Essential revisions:

1) The direction of the work (unconstrained task electrophysiology exploration) is of great interest, but the significance of this manuscript toward those goals is still not sufficiently clear. A fundamental concern is that this content seems more appropriate for a specialized methods journal. Although the authors have addressed in their rebuttal and through revisions in their paper the potential for scientific impact of this system, this does not address the point that ultimately they are working in an area of technology/methodology that has been previously reported. Claimed novelty is not correct: other groups have recorded neural data and behavior wirelessly from custom cage designs (Powell, 2017, J. Neuroscience Meth., for example). The authors also claim in the Introduction that goal-directed behaviors that involve walking have not previously been possible, but then cite Yin. Et al., 2014 and Capogrosso et al., 2016 which both involve walking behaviors and are goal (treat) directed walking behavior.

In revision, a deeper comparison to the current manuscript results is needed. First, the MaCaQue system was already published (Berger, 2019). That paper describes the behavioral collection applied to humans. The current report presents a combination of neural data collection with that behavior, and in monkey, which is interesting, but no novel behavioral or neuroscience results are provided. Please describe clearly what is unique in the system from the human version. Please present (e.g., via a table) a comparison of benchmarks across previous reports not limited to the following:

Roy and Wang, 2012; Chestek et al., 2009; Yin et al., 2014; Foster et al., 2014; Schwarz et al., 2014; Capogrosso et al., 2016.

2) The work is overly descriptive, and at some points, misleading. The neuroscience and electrophysiology leave substantial questions about the interpretation of the results and could be described more clearly. For example, the authors appear to present conclusions based on 4 neurons (out of 192) and almost no population analysis is completed (trajectories are not analysis in and of themselves). While statistical tests on those neurons is appropriate, the results do not substantiate the claims that 192 channels are recorded and convey information about goal-directed behavior.

3) Quantification of behavioral variability is also limited. Movement analysis does not describe the kinematics that are said to be collected. For example, acquisition times are given, but not variability in kinematic variables. Additional documentation of the data analyzed for the behavioral variability (Figure 3) is needed. Please report analysis of variability in movement kinematics. How many trials and behavioral sessions are presented? At what point in behavioral training are these data from? Claims of consistency/variability are difficult to assess without these details.

4) Wireless coverage rates merits more careful quantification. Wireless transmission efficacy can often vary across the cage volume due to non-uniform receiver coverage. An analysis of data loss rates that quantifies volume of the cage spanned and presence or absence of spatial trends in where data loss occurs would be significantly more compelling than aggregate data loss rates. Drop-out with strong spatial bias would hamper and complicate task analyses. This is important for the authors to quantify because their other analyses highlight that animals exhibit relatively stereotyped behaviors during tasks (Figure 3). Are the low data loss rates because the animals occupy a restricted volume of the cage during task behavior? Some quantification of coverage volume will also be valuable for reporting and demonstrating the tool's capabilities.

5) Claims for BMI utility require more careful elaboration. The authors argue their system, with real-time neural data recording and behavioral control, would be useful for BMI studies. While this is true, their system as currently designed has constraints on the possible BMI experiments that could be done which should be elaborated. Their behavioral task system, with discrete targets, would have limited applications to discrete decoding behaviors. Elaboration on specific utility and limitations/future directions is needed.

[Editors' note: further revisions were suggested prior to acceptance, as described below.]

Thank you for submitting your article "Wireless recording from unrestrained monkeys reveals motor goal encoding beyond immediate reach in frontoparietal cortex" for consideration by *eLife*. Your revised article has been reviewed by two peer reviewers, and the evaluation has been overseen by a Reviewing Editor and Joshua Gold as the Senior Editor. The following individual involved in review of your submission has agreed to reveal their identity: Samantha R Santacruz (Reviewer #1).

The reviewers have discussed the reviews with one another and the Reviewing Editor has drafted this decision to help you prepare a revised submission to clear up a few remaining points.

Summary:

The authors present a cohesive and compelling piece of work 1) demonstrating the utility of their in-cage behavioral system and wireless recordings, and 2) providing novel findings enabled by their work. The reviewers find the manuscript is greatly strengthened by the new analyses and re-framing to fully place their work in context. The combination of in-cage, untethered neural recording with well-controlled sensorimotor tasks is both powerful and a niche that has not been addressed to-date. the manuscript is suitable for publication in *eLife* subject to revisions.

Essential revisions:

1) The authors state in the rebuttal "We now added an analysis on all 12 recorded session and test for each of the 192 channels with an ANOVA if the activity is task ( = time, target distance or position) modulated and if it is target position modulated (Results section), which was not the scope of the analysis in the previous version of the manuscript. We demonstrate that there are sessions that show 192 task modulated channels and provide statistics on the population activity in the newly written part of the Results section." I believe this quoted text actually refers to: "movement. Of all twelve recorded sessions three sessions revealed task responsive activity on all 192 channels, i.e. showed at least one effect in distance, position, time or one of the interactions; across all sessions the mean number of task-responsive channels was 189 (s.d. 5 channels). Up to 179 channels were position responsive, i.e. showing at least one effect in position or one of the interactions (mean: 162, s.d. 17 channels)." It is unclear how this analysis was performed. Was "task responsive activity" determined from another ANOVA? Please clarify.

2) In the section "Premotor and parietal cortex encode movement goals beyond immediate reach". Here the authors perform SVM classification using neural spiking activity to decode reach to either a near or far target. This analysis is problematic in that they determine significant decoding accuracy by comparing performance to that of a classifier using activity prior to cue onset and also assume chance levels of 50% (based on plots in Figure 5B). It would be more appropriate to compare accuracy to shuffled data in order to determine true chance levels (particularly if the near and far trials aren't balanced) and significance. Please perform the permutation-based test of significance.

3) The authors state: "Moreover, we could decode walk-and-reach target location information from premotor and parietal cortex, but not motor cortex, during movement and even during the memory period before the movement. This suggests that premotor and parietal cortex encodes motor goals beyond immediate reach." A decoding model and encoding model are complimentary, but not the same. High decoding accuracy from the SVM simply means that the population-level neural activity during these timepoints is separable enough in a high dimensional space. This may be due to many factors. To more completely demonstrate that premotor and parietal cortex encode motor goals beyond immediate reach, an encoding model (e.g. regression or GLM) describing how neural activity co-varies with the motor goals would be valuable.

---

## [Author Response]

[Editors’ note: the authors resubmitted a revised version of the paper for consideration. What follows is the authors’ response to the first round of review.]

[…]The fundamental concern is that the behavioral apparatus alone seems more suited to a more specialized methods journal and it is not clear what kind of important scientific advance the wireless and motion capture components can support.

We are not aware of previous work studying large-scale neural network activity in freely moving monkeys trained on goal-directed naturalistic walk-and-reach movements that are temporally and spatially structured under the control of the experimenter. We provide new wireless neurophysiology and motion capture data. We show broadband intracortical recordings from 192 channel at 30 ksps from three brain areas simultaneously. Motion capture is now fully markerless and provides 3D tracking of four body-parts namely head, left wrist, elbow and shoulder. We show with an additional analysis that single-trial neural dynamics of multiple areas can be analyzed and correlated with detailed multi-joint kinematics of on unrestrained behavior. Real-time read-out of the neural data makes the setting suitable for BMI applications. Details are described below.

More clarification and quantification of the motion tracking performance is needed. The successful frame capture rate needs to be quantified over intervals of fixed duration and how this compares with reported performance of similar motion tracking methods that are not in-cage needs to be presented. Data for the second subject, Monkey L, also needs to be presented to give a second animal that would strengthen the claims of overall system performance. The inability to monitor more than 1 DOF is also considered a major limitation for a system meant ultimately to examine modulation of activity in an unconstrained animal, and this needs to be addressed.

Now, we use a different approach for markerless motion tracking based on a deep neural network. We present new data from both monkeys performing a memory-guided walk-and-reach task. The head, left wrist, elbow and shoulder are tracked in 3D during performance of the walk-and reach task. We increased the framerate to 60 Hz and quantified the successful frame capture rate after outlier rejection (97.58% – 99.95%).

The main goal of the presented research approach is to provide an experimental setting that allows studying structured and goal-oriented behavior beyond what is possible in a chair-based setup, for instance walk-and-reach behavior towards targets outside the immediate peripersonal space. We tried to make our rational clearer throughout the manuscript.

The wireless recordings also need to be analyzed to quantify in detail wireless performance for the 31 and 127 channel recording setups using standard metrics, such as BER. Since the work is done in a non-ideal, highly reflective environment, we would be satisfied if the performance was close to the worst-case reported BER value of 10^-2 (taken from Yin et al., 2014, from which the current equipment is derived).

We now use 2x 96-channel wireless headstages simultaneously to records from all 192 electrodes at 30 ksps sampling rate at the same time. Our former wireless system was not derived from Yin et al., however, our current system distinctly is. Since we do not have direct access to the digital data stream of this proprietary system, we cannot perform BER measures as shown by Yin et al. or other publications about wireless systems. Also, we do not believe that a standard BER measure is suitable to judge the relevant performance, since it is a property of the system itself and measured under optimal conditions for wireless transmission. Here, we report an experimental environment and not a wireless recording system. Thus, we report transmission stability, i.e. the fraction of time points for which we received data (around 96.68%). This is an equivalent metric to BER, in our application, as instead of measuring the quality of the wireless transmission signal itself, we measure the ability to receive the signal within our recording environment during a recording session. We are not aware of corresponding stability measures for comparable setups in which monkeys relocate themselves. This includes Yin et al., who provide data of monkeys sleeping or walking on a treadmill. In both cases, the head of the monkey, and consequently the wireless transmitter, stays approximately in the same position, which is technically less challenging.

[Editors’ note: what follows is the authors’ response to the second round of review.]

Essential revisions:1) The direction of the work (unconstrained task electrophysiology exploration) is of great interest, but the significance of this manuscript toward those goals is still not sufficiently clear. A fundamental concern is that this content seems more appropriate for a specialized methods journal. Although the authors have addressed in their rebuttal and through revisions in their paper the potential for scientific impact of this system, this does not address the point that ultimately they are working in an area of technology/methodology that has been previously reported. Claimed novelty is not correct: other groups have recorded neural data and behavior wirelessly from custom cage designs (Powell, 2017, J. Neuroscience Meth., for example). The authors also claim in the Introduction that goal-directed behaviors that involve walking have not previously been possible, but then cite Yin. et al., 2014 and Capogrosso et Al., 2016, which both involve walking behaviors and are goal (treat) directed walking behavior.

Our manuscript is not primarily about wireless transmission, it is about a conceptual approach in cognitive and sensorimotor neuroscience bearing on, among other things, wireless recordings. In this sense the study of Powell et al., 2017, which presents an RF transparent cage, i.e. equipment for wireless recording, has a very different scope. We present an experimental environment and approach for which precisely guided and quantifiable spatially and temporally instructed behavior is possible with unrestrained macaques while conducting high-bandwidth wireless recordings from multiple brain areas simultaneously. The combined features of the Reach Cage allow experimental designs that former neurophysiology studies/environments with unrestrained monkeys did not provide:

– Precise computer-controlled timing of temporal instructions (programmable color illumination of targets) to the animal and real-time control of the experimental flow based on the animals registered behavior (touch-sensitivity of the targets).

– As a consequence, investigation of motor planning is possible and demonstrated here. No other study known to us dealt with motor planning in unrestrained monkeys.

– Multiple distributed, light identifiable and instructable reach goals that are independent of the food source (important in cognitive neuroscience) and can be configured variably in number and position.

– As a consequence, a defined starting position with the ability to pause the animal’s movement allows to provide visual cues at a defined locations relative to the body, necessary for motor-cognitive aspects of sensorimotor science. For example, trained goal directed behavior led to a distinct number of behavioral categories (here movements to 8 different targets) and repetitive trials as previously possible only in more conventional chair-seated paradigms (see also response to comment 3). We are not aware of a study showing that pure training of monkeys without any physical restraint leads to such low trial-to-trail variability in freely moving animals.

We have revised the Introduction and Discussion to make the idea behind our study more clear. We have added an overview of previous studies, Supplementary file 1, as suggested below.

As requested by the editor’s summary, we provide more detail on our neurophysiology data as a proof-of-concept. For this, we replaced the last subsection in the Results section with a new analysis of the neurophysiology data that shows that premotor and parietal cortex encode motor goals beyond immediate reach during planning and locomotion. We are not aware of existing experimental environments or published data that would show this or would have been able to show this.

In revision, a deeper comparison to the current manuscript results is needed. First, the MaCaQue system was already published (Berger, 2019). That paper describes the behavioral collection applied to humans. The current report presents a combination of neural data collection with that behavior, and in monkey, which is interesting, but no novel behavioral or neuroscience results are provided. Please describe clearly what is unique in the system from the human version.

We are surprised by this argument given *eLife*’s support for preprints and find it in fact a highly problematic argument. The MaCaQuE system was first published as the preprint of this manuscript. In Berger et al., 2019, we cited the preprint accordingly. We will cite this manuscript, not Bergeret al.,2019, when referring to the MaCaQuE System, since only here we present an extensive description of the MaCaQuE system. Since the current manuscript was already in the pipeline at *eLife* and available as preprint, we think this is the only appropriate way of dealing with such situation. Since the publication history can easily be tracked in the archive system, the genesis is transparent. Regarding novelty, Berger et al., 2019, is all about a human psychophysics result showing congruency effects in the interference of cross-modal sensory integration with spatial distractor stimuli, not about the MaCaQuE system and not about non-human primates, as evident from the paper. The novelty here is explained in our response to the above comment, for which MaCaQuE is an important but not the sole component. Specifically, the novelty here is directed towards studies with non-human primates. The current and previous studies have a completely independent scope.

We find the argument highly problematic since it effectively means that throughout the review process, during which authors – like in our case – invest massively to follow reviewer requests for significant modifications and sometimes have to wait many months to get feedback on their submission, authors would not be allowed to make reference to their published preprints in other studies. This defeats the purpose of preprints and in our view contradicts *eLife*’s policy.

Please present (e.g., via a table) a comparison of benchmarks across previous reports not limited to the following:Roy and Wang, 2012; Chestek et al., 2009; Yin et al., 2014; Foster et al., 2014; Schwarz et al., 2014; Capogrosso et al., 2016.

We present this table as Supplementary file 1.

2) The work is overly descriptive, and at some points, misleading. The neuroscience and electrophysiology leave substantial questions about the interpretation of the results and could be described more clearly. For example, the authors appear to present conclusions based on 4 neurons (out of 192) and almost no population analysis is completed (trajectories are not analysis in and of themselves). While statistical tests on those neurons is appropriate, the results do not substantiate the claims that 192 channels are recorded and convey information about goal-directed behavior.

Since the focus of the manuscript, based on the reviewer suggestions, has now partly shifted towards more neuroscientific content, we also adapted the style and it should be less descriptive. The methods-oriented parts, which were originally more in the foreground since we submitted to the “Tools and Resources” rubric of the journal, remain descriptive. The example units are shown to demonstrate that it is possible to record clearly modulated, well-isolated single-unit activity in much detail, with modulations of the neural responses following closely the precisely controlled different stages of the trials over time and at the same time being selective for the spatial parameters of the task. We now added an analysis on all 12 recorded session and test for each of the 192 channels with an ANOVA if the activity is task ( = time, target distance or position) modulated and if it is target position modulated (Results section), which was not the scope of the analysis in the previous version of the manuscript. We demonstrate that there are sessions that show 192 task modulated channels and provide statistics on the population activity in the newly written part of the Results section.

3) Quantification of behavioral variability is also limited. Movement analysis does not describe the kinematics that are said to be collected. For example, acquisition times are given, but not variability in kinematic variables. Additional documentation of the data analyzed for the behavioral variability (Figure 3) is needed. Please report analysis of variability in movement kinematics. How many trials and behavioral sessions are presented? At what point in behavioral training are these data from? Claims of consistency/variability are difficult to assess without these details.

We do report an analysis of variability of movement kinematics (Figure 3B and third paragraph in this subsection). This is the across time points averaged Euclidean distance for each trial to a trial-averaged trajectory. We show that trials deviate only few centimeters from the averaged trajectory. Even for the most variable marker (wrist) the averaged Euclidean distance is below 67 mm for 75% of the trials.

We now added more sessions to this analysis. Now the analysis is based on 2 (monkey K) and 3 sessions (monkey L) with a total of 469 and 872, respectively, successful trials. We added this information in the Results section and Figure 3.

All data presented in the manuscript is after behavioral training. We added this information in the Materials and methods section.

4) Wireless coverage rates merits more careful quantification. Wireless transmission efficacy can often vary across the cage volume due to non-uniform receiver coverage. An analysis of data loss rates that quantifies volume of the cage spanned and presence or absence of spatial trends in where data loss occurs would be significantly more compelling than aggregate data loss rates. Drop-out with strong spatial bias would hamper and complicate task analyses. This is important for the authors to quantify because their other analyses highlight that animals exhibit relatively stereotyped behaviors during tasks (Figure 3).

We added an analysis of data loss rates across targets (Results section and Figure 4—figure supplement 1). While the raw data shows some dependency of data loss rates on the target position in the range of few percent, we did not observe significant differences between trials towards different target positions in the left-right dimension after applying our 5% criterion for rejecting poor trials. We do see a difference between reach and walk-and-reach movements; however, this does not affect our results since these quantify left-right selectivity for reach and walk-and-reach movements independently.

Are the low data loss rates because the animals occupy a restricted volume of the cage during task behavior? Some quantification of coverage volume will also be valuable for reporting and demonstrating the tool's capabilities.

Since monkeys performed a spatially and temporally highly structured behavior during the recording sessions, we cannot draw conclusions on the coverage beyond the space covered in Figure 4—figure supplement 1.

5) Claims for BMI utility require more careful elaboration. The authors argue their system, with real-time neural data recording and behavioral control, would be useful for BMI studies. While this is true, their system as currently designed has constraints on the possible BMI experiments that could be done which should be elaborated. Their behavioral task system, with discrete targets, would have limited applications to discrete decoding behaviors. Elaboration on specific utility and limitations/future directions is needed.

We thank the reviewer for pointing this out. We used the opportunity to rewrite the corresponding Discussion section. On the one hand, we are specifically interested in the control of discrete events, e.g. for smart home applications, for which spatial selectivity/invariance regarding the subjects position in space is important. On the other hand, robustness of prosthetic control during performance of other movements in parallel, like walking, is critical for applications in partially impaired patients (e.g. arm amputees). The new neural analyses show that target location information can be decoded during walking, as a first step to show that information for BMIs can be harnessed from the neural activity recorded in the Reach Cage. We now elaborate on this in the Discussion section, but since BMI application is not the main focus of this study, we removed the paragraph from the Introduction.

[Editors' note: further revisions were suggested prior to acceptance, as described below.]

Essential revisions:1) The authors state in the rebuttal "We now added an analysis on all 12 recorded session and test for each of the 192 channels with an ANOVA if the activity is task ( = time, target distance or position) modulated and if it is target position modulated (Results section), which was not the scope of the analysis in the previous version of the manuscript. We demonstrate that there are sessions that show 192 task modulated channels and provide statistics on the population activity in the newly written part of the Results section." I believe this quoted text actually refers to: "movement. Of all twelve recorded sessions three sessions revealed task responsive activity on all 192 channels, i.e. showed at least one effect in distance, position, time or one of the interactions; across all sessions the mean number of task-responsive channels was 189 (s.d. 5 channels). Up to 179 channels were position responsive, i.e. showing at least one effect in position or one of the interactions (mean: 162, s.d. 17 channels)." It is unclear how this analysis was performed. Was "task responsive activity" determined from another ANOVA? Please clarify.

We understand why this paragraph is unclear and added an explanatory sentence: “We performed the same ANOVA for the activity in each channel of all twelve recorded sessions.“

2) In the section "Premotor and parietal cortex encode movement goals beyond immediate reach". Here the authors perform SVM classification using neural spiking activity to decode reach to either a near or far target. This analysis is problematic in that they determine significant decoding accuracy by comparing performance to that of a classifier using activity prior to cue onset and also assume chance levels of 50% (based on plots in Figure 5B). It would be more appropriate to compare accuracy to shuffled data in order to determine true chance levels (particularly if the near and far trials aren't balanced) and significance. Please perform the permutation-based test of significance.

We replaced the statistics on the decoding analysis with a permutation-based test of significance for our main analysis (Results, Figure 5 and Figure 5—source data 1) and for our supplementary analysis (Results, Figure 5—figure supplement 1 and Figure 5—source data 3). The permutation-based test revealed the same results as the test in the previous version of the manuscript, apart from now one additional significant decoding accuracy in M1 of monkey L during movement (but not within the memory period) of walk-and-reach movements. All previously reported effects are still valid. This minimally different result does not change the conclusions of our study. We revised the Materials and methods section accordingly.

3) The authors state: "Moreover, we could decode walk-and-reach target location information from premotor and parietal cortex, but not motor cortex, during movement and even during the memory period before the movement. This suggests that premotor and parietal cortex encodes motor goals beyond immediate reach." A decoding model and encoding model are complimentary, but not the same. High decoding accuracy from the SVM simply means that the population-level neural activity during these timepoints is separable enough in a high dimensional space. This may be due to many factors. To more completely demonstrate that premotor and parietal cortex encode motor goals beyond immediate reach, an encoding model (e.g. regression or GLM) describing how neural activity co-varies with the motor goals would be valuable.

We added an encoding (“tuning”) analysis. As we are interested in population and not single unit effects, we calculated the population average of the firing rate modulation, i.e. absolute difference between average firing rate for left and right trials and then averaged over all units after aligning to the preferred side (“population tuning”). As for the decoding analysis, we calculated a permutation-based test to test for significant encoding (Figure 5 and Figure 5—source data 2) or significant differences between passage and open trials (Figure 5—figure supplement 1 and Figure 5—source data 4). We describe this accordingly in the Results and Materials and methods. The results for the encoding analysis do not differ from the decoding analyses, hence the conclusions of the study remain the same.